# Structure of a distinct β-barrel assembly machinery complex in the Bacteroidota

Augustinas Silale [1,5], Mariusz Madej[2,5] ✉, Katarzyna Mikruta [2,3], Andrew M. Frey [1], Adam J. Hart [1], Arnaud Baslé[1], Carsten Scavenius [4], Jan J. Enghild [4], Matthias Trost [1], Robert P. Hirt[1] & Bert van den Berg [1] ✉

The Gram-negative β-barrel assembly machinery (BAM) complex catalyses the folding and membrane insertion of newly synthesized β-barrel outer membrane proteins. The BAM is structurally conserved, but most studies have focused on Gammaproteobacteria. Here, using single-particle cryogenic electron microscopy, quantitative proteomics and functional assays, we show that the BAM complex is distinct within the Bacteroidota. Cryogenic electron microscopy structures of BAM complexes from the human gut symbiont *Bacteroides thetaiotaomicron* (3.3 Å) and the human oral pathogen *Porphyromonas gingivalis* (3.2 Å) show similar, seven-component complexes of ~325 kDa. The complexes are mostly extracellular and comprise canonical BamA and BamD; an integral, essential outer membrane protein, BamG, that associates with BamA; and four surface-exposed lipoproteins: BamH–K. Absent from the BAM in Pseudomonadota, BamG–K form a large, extracellular dome that may confer additional functionality to enable the folding and assembly of β-barrel–surface-exposed lipoprotein complexes that are a hallmark of the Bacteroidota. Our findings develop our understanding of fundamental biological processes in an important bacterial phylum.

The β-barrel assembly machinery (BAM) complex is a conserved, multiprotein machine that mediates the folding and membrane insertion of newly synthesized β-barrel outer membrane (OM) proteins. The BAM complex structure and mechanism of action have been studied in detail in Gammaproteobacteria, particularly *Escherichia coli*. These studies have shown that the archetypal BAM complex is ~200 kDa in size and consists of an essential, 16-stranded integral β-barrel OM protein (OMP) named BamA, which has five periplasmic polypeptide-transport-associated (POTRA) domains that are decorated in the periplasmic space by the four lipoproteins BamB–E[1–6]. BamA is an atypical β-barrel in which the first and last β-strands can separate to form a lateral gate that is open towards the OM. Recent studies show that the 'uncovered' first BamA strand acts as a template to fold nascent

OMPs via a 'hybrid-barrel budding' mechanism[7–9], probably facilitated by local OM destabilization and thinning[10]. The BamB–E lipoproteins probably increase the efficiency of this process in ways that are not yet clear despite two decades of study[11,12]. Except for BamD, they are dispensable for cell viability in the absence of suppressor mutations[13]. Delivery of unfolded substrates to the BAM complex is mediated by several periplasmic chaperones such as SurA and Skp, which function in distinct ways[14–17]. Phylogenetic analysis suggests only minor variations for BAM complex composition across bacterial phyla, caused by variable presence of the non-essential components BamB, BamC and BamE[18]. In addition to β-barrel integral OMPs, the OM contains lipoproteins. In Pseudomonadota, most of these are targeted to and inserted into the inner leaflet of the OM via the Lol pathway, resulting

[1]Biosciences Institute, Faculty of Medical Sciences, Newcastle University, Newcastle upon Tyne, UK. [2]Department of Microbiology, Faculty of Biochemistry, Biophysics and Biotechnology, Jagiellonian University, Krakow, Poland. [3]Doctoral School of Exact and Natural Sciences, Jagiellonian University, Krakow, Poland. [4]Interdisciplinary Nanoscience Center (iNANO), Aarhus University, Aarhus, Denmark. [5]These authors contributed equally: Augustinas Silale, Mariusz Madej. ✉e-mail: mariusz.madej@uj.edu.pl; bert.van-den-berg@newcastle.ac.uk

in periplasmic exposure[19]. In addition to these 'classical' lipoproteins, surface lipoproteins (SLPs) also exist that, after being targeted to the OM, need to be flipped for surface exposure[20]. Structurally and mechanistically, little is known about SLP flipping, although in Pseudomonadota the OMP SLP assembly modulator (Slam) is known to be involved in a BAM-independent fashion[21,22]. Importantly, the numbers of SLPs in Pseudomonadota are generally low (~10 in *E. coli*), and complexes between SLPs and β-barrel OMPs, exemplified by the TbpAB iron piracy system of pathogenic *Neisseria*[23], are uncommon. By contrast, stable SLP–barrel complexes are ubiquitous and abundant within the Bacteroidota, a large Gram-negative bacterial phylum that is widely distributed within the environment (soil, seawater) and in animal microbiomes. The model human gut symbiont *Bacteroides thetaiotaomicron* has an estimated 400–450 SLPs, based on the presence of the Bacteroidota lipoprotein export signal[24,25] in a large subset of *B. thetaiotaomicron* lipoproteins. Many of these SLPs are present in complexes with large TonB-dependent transporters (TBDTs) that mediate the uptake of a wide range of dietary and host glycans, vitamin $B_{12}$ and iron-siderophores[26,27]. These SLP–barrel complexes are likely to be key for the success of the genus *Bacteroides* in the human gut, but how the SLPs are flipped across the OM and how the SLP–barrel complexes are subsequently assembled are completely unclear.

Here we report the discovery of a distinct BAM complex that is widespread within the Bacteroidota. Cryogenic electron microscopy (cryo-EM) structures of the BAM complexes from the gut commensal *B. thetaiotaomicron* and the oral pathogen *Porphyromonas gingivalis* are similar and reveal a 7-protein, ~325-kDa complex with most of its mass on the extracellular side. Remarkably, besides BamA and BamD subunits, Bacteroidetal BAM contains a 14-stranded integral OMP that is intimately associated with BamA. We name this subunit BamG and show that it is a homologue of the *E. coli* long-chain fatty acid transporter FadL. Repeated failure to delete BamG-encoding genes both in *B. thetaiotaomicron* and in *P. gingivalis* suggests it is essential. The remaining four components, named BamH–K, are all SLPs and form an extracellular dome-like structure over BamA that may provide additional functionality to enable the folding and assembly of the many β-barrel–SLP complexes that are a hallmark of the Bacteroidota.

## Results

### BtBamA co-purifies with several OMPs of unknown function

We reasoned, based on the need to assemble large numbers of structurally divergent SLP–barrel complexes in *B. thetaiotaomicron*, that a BAM complex of this organism might have interesting properties. To test this hypothesis, we tagged the chromosomal copy of the gene encoding BtBamA (*bt3725*) with an N-terminal His₇ tag and performed pulldowns from *B. thetaiotaomicron* membrane extracts in dodecyl-maltoside (DDM). Several proteins co-purified with BtBamA_{his}, and their identities were established via peptide fingerprinting of pooled size exclusion chromatography (SEC) peak fractions (Fig. 1a) and quantitative proteomics on immobilized metal affinity chromatography (IMAC) elutions (Fig. 1b, Extended Data Fig. 1 and Supplementary Table 1). Unexpectedly, besides BamD (Bt0573) and SurA (Bt3848) that constitute a 'minimal' Proteobacterial BAM system, BamA_{his} also co-purifies with an integral β-barrel OMP that we name BamG (Bt4367), given that a BamF protein was previously described in Alphaproteobacteria[28]. In addition to BamG, BamA co-purifies with two lipoproteins of unknown function, Bt4306 (BamH) and Bt3727 (BamI), and two peptidyl-prolyl isomerases (PPIs), Bt3612 (BamJ) and Bt3949 (BamK). Inspection of the sequences shows that BamH–K all have at least two acidic residues within the first six residues after the lipid anchor cysteine, suggesting that they are SLPs[24]. The pulldown data support bioinformatics analyses that suggest Bacteroidota lack homologues of BamC and BamE. Whether any Bacteroidota have BamB is not clear, but our data strongly suggest that *B. thetaiotaomicron* does not. AlphaFold[29] predictions of the five proteins (Extended Data Fig. 2) did not give obvious clues

about a function within a putative BAM complex. Interestingly, BamG is one of the four *B. thetaiotaomicron* paralogs of the long-chain fatty acid transporter FadL from *E. coli*[30,31], and structurally similar to Bt1785 (Extended Data Fig. 2a,b). BamG (FadL/Toluene_X) and the PPI BamJ (FKBP_C) are widespread (Extended Data Fig. 3), whereas BamH, BamI and the PPI BamK appear to be confined to the Bacteroidota. Of these, BamH is widespread, while BamI and BamK are largely restricted to the Bacteroidia class (Extended Data Fig. 3). *B. thetaiotaomicron* has one paralog of BamH, Bt1786 (BamH2), which is ~100 residues shorter than BamH but has a similar predicted structure (Extended Data Fig. 2b).

### The bulk of the BtBAM complex is extracellular

To obtain more information on the relationship of the co-purified proteins with BtBamA, we collected single-particle cryo-EM data on a sample in DDM purified from ~10 l of *B. thetaiotaomicron* grown in rich medium. Maps to ~3.3 Å resolution were obtained (Supplementary Fig. 1a–d and Methods) that allowed confident manual docking of AF2 models and subsequent refinement (Fig. 2a,b and Supplementary Table 2). All the abundant (that is, visible as a clear band in sodium dodecyl sulfate-polyacrylamide gel electrophoresis (SDS-PAGE) after size exclusion chromatography) co-purified proteins except Bt0502 (Fig. 1a) are present in the cryo-EM volumes, each with one copy to generate a large complex of ~325 kDa, which we term BtBAM (Fig. 2a,b). In striking difference to the BAM from Pseudomonadota, most of the BtBAM complex is extracellular owing to the four newly discovered SLPs. Three-dimensional variability analysis[32] showed heterogeneity on the extracellular and periplasmic sides of the micelle (Supplementary Video 1). Further classification and refinement revealed the presence of two particle classes in the dataset (Fig. 2a and Supplementary Fig. 1a). Class 1 lacks density for the periplasmic components but has good density for all extracellular proteins, whereas class 2 has modest resolution for BamA POTRA domains 4 and 5 and for part of BamD, but with poorly resolved extracellular areas (Supplementary Fig. 1e,f). Given that models derived from buildable areas in both maps are very similar with only small (maximum ~1 Å for Cα atoms) rigid body movements of the extracellular domain, we present only one, the 'hybrid' structural model. The dataset does not contain any particle populations of sub-complexes, as suggested by heterogeneous refinement and 3D variability analysis results (Supplementary Fig. 1a and Supplementary Video 1).

The 14-stranded barrel of the integral membrane component BamG occupies a key position within BtBAM. It forms a tight complex with the back of BamA (defined as being located opposite the BamA lateral gate), with an interface area of ~1,560 Å² as analysed via PISA[33], and binds Bt4306 (renamed BamH) via its extracellular loops (interface area ~1,720 Å²) (Fig. 2b). The latter interaction is somewhat reminiscent of the proposed role of PorV, also a FadL homologue, as a substrate-binding and shuttling protein in the type 9 secretion system (T9SS)[34]. However, given the intimate association of BamG with both BamA and BamH, we consider it unlikely that BamG has a PorV-like function in BtBAM. A feature that clearly distinguishes BamG from FadL channels with a known transport role is an extension of the N-terminus of ~10 residues, which protrudes through the lateral opening in the barrel wall that normally is part of the FadL transport pathway (Extended Data Fig. 4a)[35,36]. Interestingly, a ConSurf[37] analysis shows that the extended BamG N-terminus is highly conserved, but the extracellular loops interacting with BamH are not (Extended Data Fig. 4b). The extended N-terminus is also present in Bt1785 (BamG2; Extended Data Fig. 4a), but a detailed phylogenetic analysis shows that it, and the additional BamGs encoded by *Bacteroides* spp. with more than one *bamG* gene, are distinct from BtBamG (Extended Data Fig. 4c,d). BlastP searches against the 1,448 complete and annotated Bacteroidota genomes in the RefSeq database (National Center for Biotechnology Information, NCBI) using BtBamG as query identified one to three significant hits per genome in all but one free-living

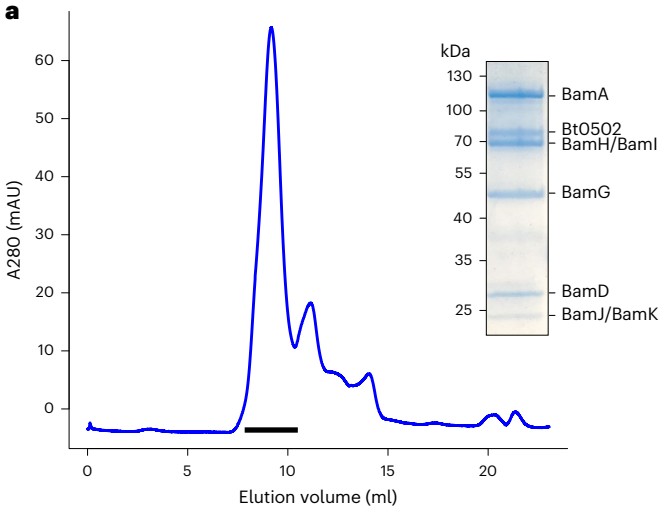

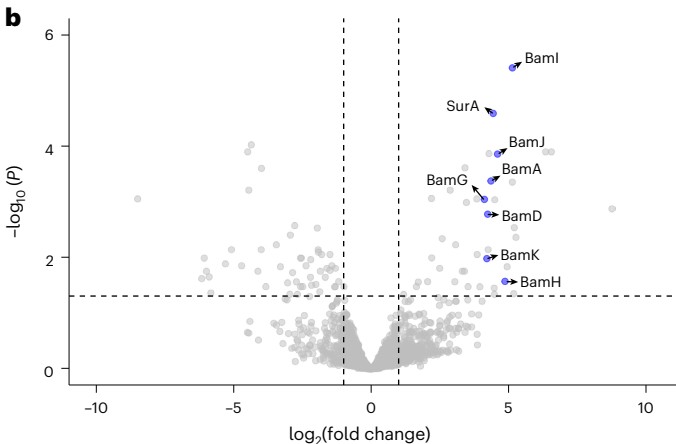

**Fig. 1 | Identification of BtBAM components. a**, Representative size-exclusion chromatography trace of a BtBamA_his pulldown (*n* = 5). Fractions denoted by the black horizontal bar were pooled, concentrated and analysed by 12% SDS-PAGE (inset). Individual bands were cut out and identified by peptide fingerprinting. Bt0502 is a background contaminant (Supplementary Table 1). **b**, Volcano plot showing enrichment of proteins in pulldowns from *B. thetaiotaomicron bamA_his* versus wild-type cells (*n* = 3 biological replicates for both conditions). The vertical dashed lines represent log$_2$(fold change) = ±1. Proteins above the horizontal dashed line have an adjusted *P* < 0.05. Statistical significance was inferred by two-sided *t*-test with Benjamini–Hochberg correction for multiple comparisons.

Bacteroidota (Supplementary Table 3 and Supplementary Discussion). No significant hits were found in any endosymbiont Bacteroidota. This is consistent with an important role played by BamG and its functional partners in processing cell surface proteins, the repertoire of which is typically dramatically reduced within the small genomes of endosymbiotic bacteria, which maintain only basic functionalities such as the biosynthesis of essential amino acids[38]. Indeed, repeated attempts to delete BtBamG failed, suggesting that it is essential. By contrast, BamG2 could be deleted readily.

The crescent-shaped BamH subunit is positioned over the BamA barrel and forms an arch-like structure approximately 60 Å above the OM surface (Extended Data Fig. 5a). BamH interacts with BamI and BamJ on one side and with BamK on the other. Based on its structure and the conservation of the protein core, BamH probably functions as a scaffold. The highly conserved N-terminus of BamH is wedged between two strands of the BamG barrel and interacts with the conserved BamG N-terminus, and the tri-acylated BamH lipid anchor is positioned at the BamAG interface (Extended Data Fig. 5a,b). Together, BamG–K form

a striking, half dome-shaped structure in the extracellular space, with its apex almost exactly above the lateral opening of BamA (Fig. 2d). Of the remaining three components, BamI consists of two domains. The small N-terminal domain is conserved and interacts with the BamJ PPI, whereas the less-conserved C-terminal domain contacts BamH and adopts a six-bladed β-propeller fold, with each blade consisting of three to four β-strands (Extended Data Fig. 5c). Neither domain provides clues as to the potential role of BamI within the complex. The *E. coli* BAM complex also contains a β-propeller lipoprotein, BamB, facing the periplasm, but it is unclear whether the functions of the β-propellers are analogous or distinct in the two organisms. The conservation analysis of both BamJ and BamK indicates that their putative active sites are directed towards the interior of the BAM dome (Extended Data Fig. 5d,e). A total of 17 glycosylation sites were observed in the cryo-EM density (Extended Data Fig. 5f). Most of these are extracellular, with only one located on the periplasmic side on BamG. BamH–K all have glycosylation sites, with by far the most (10) present on BamI. All sites have the consensus D-T/S-hydrophobic motif for O-glycosylation in Bacteroidota[39]. It is unclear whether these post-translational modifications have any functional relevance.

Similarly to Proteobacterial BAM complex structures, the cryo-EM maps suggest considerable heterogeneity and/or mobility within BamA and BamD. Density for BamA POTRA domains 1–3 and for the N-terminal region of BamD is missing (residues 18–71), and the visible periplasmic components have a relatively low resolution (Supplementary Fig. 1f). Likewise, density for β1 and β2 strands of BamA is poor, suggesting heterogeneity at the lateral gate and a loose association of β1 and β16 strands (Fig. 2c and Supplementary Fig. 1e,f). The *E. coli* BamA β-barrel seam has been observed in various degrees of opening, and the seam is disordered in our BtBamA structure[1,40,41]. We also noticed increased disorder of the detergent micelle at the BamA β-barrel seam (Supplementary Fig. 2), consistent with results from molecular dynamics simulations showing membrane thinning at the *E. coli* BamA β-barrel gate[2,42].

With regard to the BamA extracellular loops, there are notable differences between BtBamA and *E. coli* BamA (EcBamA). BtBamA has a striking extracellular loop (EL) 3 between the β5 and β6 strands, which contains 13 tyrosine residues and probably projects inside the dome formed by the SLPs (Extended Data Fig. 6a–d). Sequence alignment of Bacteroidota BamA homologues indicates that EL3 is also enriched in glycine and asparagine residues (Extended Data Fig. 6e). There was no cryo-EM density observed for this 38-residue loop, which suggests that it samples a range of different conformations. The function of the tyrosine-rich EL3 is unclear, but its flexibility implies that it does not bind to the other BtBAM components. The β-barrel lumen in both BtBamA and EcBamA is occluded mainly by EL6, but these loops differ in length and structure, with BtBamA EL6 being much shorter (Extended Data Fig. 6a–c). Both proteins have a short, partially α-helical EL corresponding to EL5 in BtBamA and EL4 in EcBamA.

## The BtBAM architecture is conserved in *P. gingivalis*

To obtain more information on the structural conservation of the Bacteroidota BAM complex, we turned to *P. gingivalis* strain ATCC 33277, a prominent, well-studied oral pathogen belonging to the family Porphyromonadaceae. Compared with the large genomes of *Bacteroides* spp. (*B. thetaiotaomicron*; ~6.2 Mb, ~4,800 open reading frames), *P. gingivalis* has a much smaller genome (~2,100 open reading frames), making it an interesting organism to assess the conservation of the BAM complex. Tagging PgBamA at the N-terminus led to low yields of intact complex owing to proteolytic degradation of the periplasmic domain, presumably by the ubiquitous gingipain surface proteases characteristic of *P. gingivalis*[43]. A tag added to the C-terminus of PgBamH (PGN_1735) allowed us to purify the intact complex and determine its cryo-EM structure at ~3.2 Å resolution (Fig. 3a,b, Supplementary Fig. 3 and Supplementary Table 2). The composition and architecture of the PgBAM complex is very similar to that of *B. thetaiotaomicron*, with a

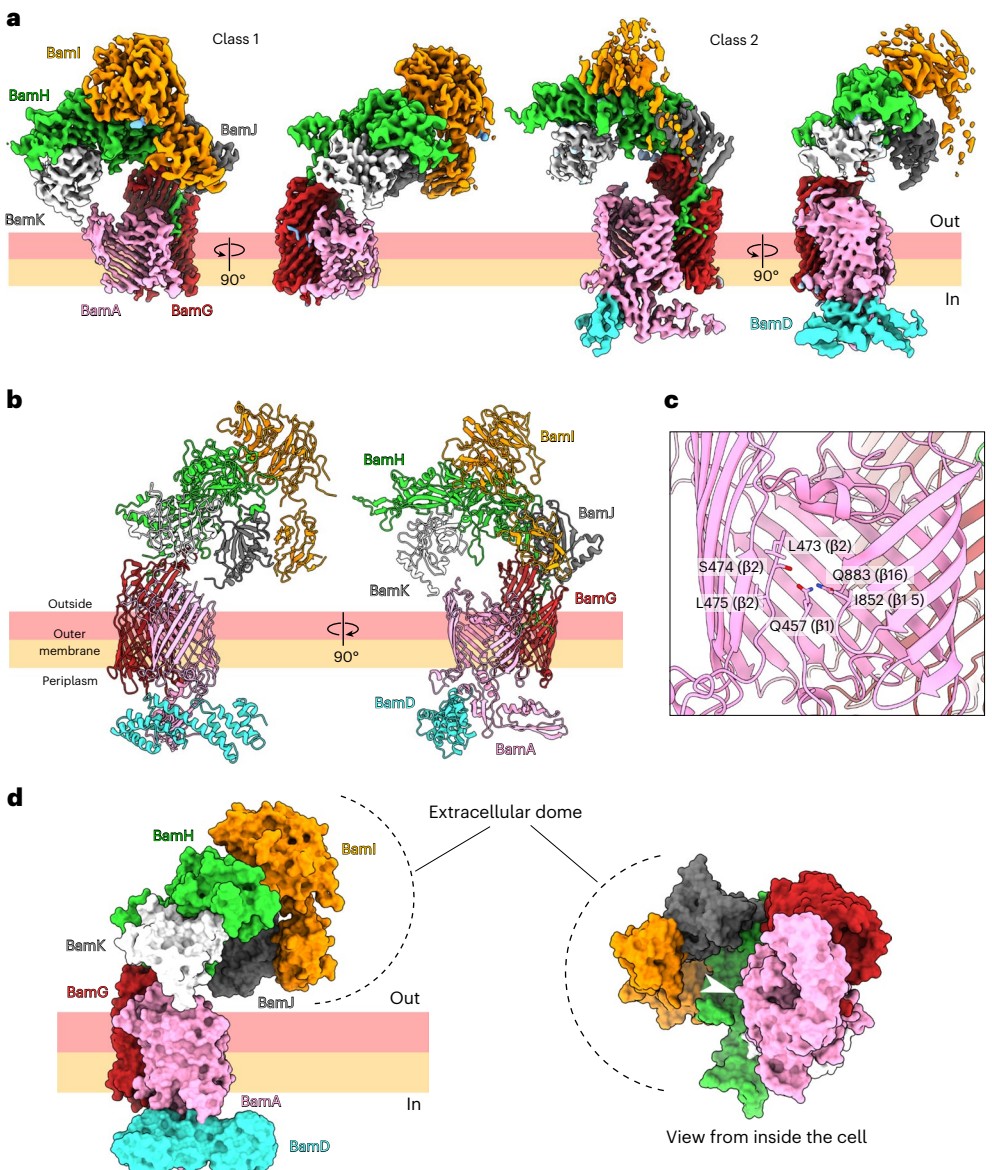

**Fig. 2 | Single-particle cryo-EM structure of BtBAM. a**, Refined cryo-EM density maps of the final two 3D classes. **b**, Model built into the cryo-EM maps. The SLP models were refined against the class 1 density in which the extracellular region was better resolved. The BamADG models were refined against class 2, in which BamA and BamD density was better resolved. The refined SLP models were rigid body docked into the class 2 map, resulting in the final model. **c**, Close-up view of the BamA β-barrel seam. Residues 458–472, comprising most of the β1 strand, extracellular loop 1 and part of the β2 strand, are not resolved in the cryo-EM maps. **d**, The SLPs form an extracellular dome, which partially encloses a ~90,000-Å³ volume above the OM. The white arrowhead in the right panel indicates the lateral opening of the BamA β-barrel, with BamD omitted for clarity.

pronounced extracellular dome formed by BamG–K. Only very weak, not modelled, density was observed for the periplasmic components, possibly owing to clipping by gingipains, similar to what was observed in the isolation of tagged PgBamA (Extended Data Fig. 7a). All components of PgBAM and BtBAM are similar in size except for BamH, which is 88 residues longer in *B. thetaiotaomicron*. Structural alignments of individual BtBAM and PgBAM components show high structural similarity for BamAGHIJ (Extended Data Fig. 8). The highest Cα-Cα root mean square deviation (RMSD) of 6.3 Å was observed for BamI, although both PgBamI and BtBamI consist of an N-terminal Ig-like domain and a C-terminal six-bladed β-propeller. The extracellular dome of PgBAM is much more compact than that of BtBAM, as indicated by the distance between the BamJ and BamK PPIs (Fig. 3c,d and Supplementary Video 2). A total of 12 glycosylation sites are present within PgBAM, mostly located on BamH (7 sites) (Extended Data Fig. 7b). The tri-acylated PgBamH lipid anchor is clearly defined and, like in *B. thetaiotaomicron*,

positioned at the BamAG interface (Extended Data Fig. 7c). EL3 in *P. gingivalis* BamA is 39 residues long and contains 12 tyrosines. Similar to BtBamA, there is no cryo-EM density for this loop in PgBamA.

## BAM SLPs are important for growth and OM integrity

As it was not possible to generate a *bamG* deletion strain, we constructed a *B. thetaiotaomicron bamH* deletion strain to investigate its function. Given its position with the BAM complex, we hypothesized that BamH removal might result in the loss of all SLP components (BamH–K). Indeed, BtBamA_His pulldowns from a Δ*bamH* background confirmed this notion (Fig. 4a,b, Extended Data Fig. 1 and Supplementary Table 1). Importantly, the absolute and relative levels of the BamADG core remained similar (Fig. 4b and Supplementary Table 1), indicating that we removed the extracellular BAM dome without affecting the core BAM components. The Δ*bamH* strain showed growth defects in rich medium and minimal medium with both

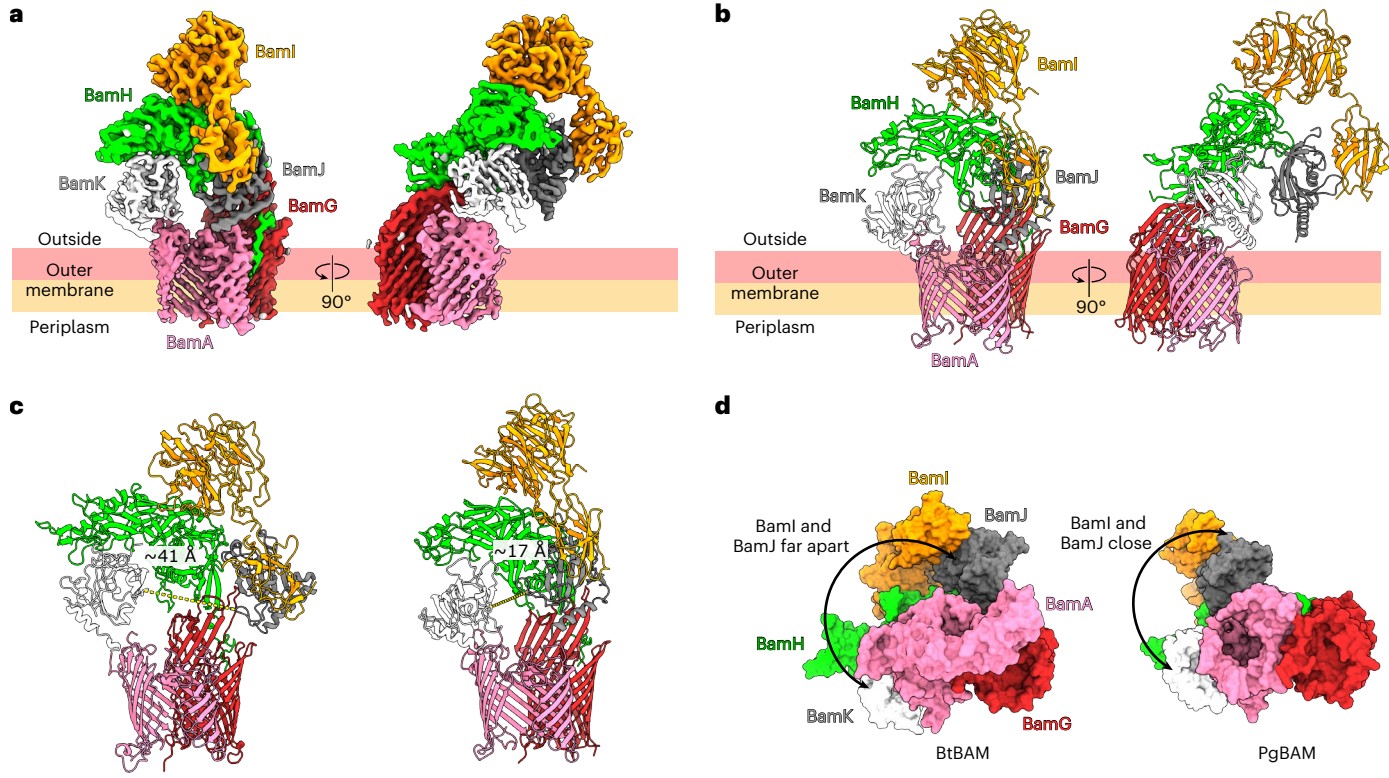

**Fig. 3 | *P. gingivalis* BAM has the same architecture as BtBAM. a**, Cryo-EM map of the PgBAM class without periplasmic density, coloured by component. **b**, PgBAM model. The colouring scheme is the same as for BtBAM. **c**, The BtBAM extracellular dome is more open than observed in the PgBAM cryo-EM structure, as shown by the distance between the nearest BamJ and BamK loops (yellow dashed line). Views were generated from a superposition on BamA. **d**, View of the BtBAM and PgBAM complexes from the cytoplasmic side. BtBamD is not shown for clarity. Views were generated from a superposition on BamA.

monosaccharides and polysaccharides as carbon sources (Fig. 4c). The deletion of *bamI* resulted in an intermediate phenotype (Supplementary Fig. 4), suggesting that the defect observed in the Δ*bamH* strain is due to a cumulative loss of BamI, BamJ, BamK and possibly BamH itself. When *B. thetaiotaomicron* was cultured in the presence of the OM stressors SDS and sodium deoxycholate, we observed a severe decrease in OM stress tolerance in the Δ*bamH* strain compared with the wild type (Fig. 4d). Thus, our data show that BamH is important in vitro, but not essential. Alternatively, its paralogue, Bt1786 (BamH2), might be able to partially complement the lack of BtBamH. BamH2 is in a putative operon with BtBamG2, and the BtBamG2H2 complex model predicted by AF2 is similar to the BamGH unit in our BtBAM cryo-EM structure (Supplementary Fig. 5). However, we have not been able to detect BamG2 or BamH2 in our proteomics data, including those for Δ*bamH* strains. In addition, a *bamH* and *bamH2* double deletion strain has a similar stress phenotype as the *bamH* deletion strain (Extended Data Fig. 9), arguing against the existence of an alternative BtBAM complex involving BtBamG2H2 that could rescue phenotypes in the Δ*bamH* strain.

For *P. gingivalis*, growth of *bamH*, *bamI* and *bamK* deletion strains in rich medium was only slightly reduced (Fig. 4e). However, we observed a drastic growth reduction in minimal medium (Dulbecco's Modified Eagle Medium (DMEM) containing 1% bovine serum albumin (BSA) as carbon and nitrogen source) when supplemented with standard concentrations of vitamin K (0.5 mg l⁻¹) and haemin (5 mg l⁻¹) (Fig. 4e; DMEM + BSA normal). Growth was further compromised when the concentration of supplements was decreased tenfold (Fig. 4e; DMEM + BSA low). Notably, the *bamI* and *bamK* mutants remained viable after 48 h, whereas the *bamH* mutant was dead shortly after inoculation. Moreover, the growth defect in minimal medium for PgBAM mutants was much more pronounced than for mutants of the RagAB

OM β-barrel–SLP complex involved in peptide uptake[44]. This suggested that the effect of these mutations on growth might result from a cumulative impairment of SLP–barrel complex assembly in the OM.

### Deletion of *bamH* affects cell surface structure biogenesis in *B. thetaiotaomicron*

We reasoned that deletion of *bamH* may have resulted in changes to the proteome that were responsible for the OM integrity defects. To compare the relative abundance of membrane proteins in *B. thetaiotaomicron* wild type and the Δ*bamH* strain, we used quantitative proteomics using intensity-based absolute quantification[45] on total membrane fractions. A total of 131 proteins were less abundant ($\log_2$(fold change) < −0.58; $P$ < 0.05) in the Δ*bamH* strain compared with wild-type *B. thetaiotaomicron* (Fig. 5a, Supplementary Table 4 and Supplementary Discussion). Pulldown experiments showed that deletion of *bamH* leads to loss of all SLPs from the BtBAM complex while BamADG levels remain similar (Fig. 4a,b and Supplementary Table 1), but the total membrane proteomics show that only the BamH and BamK SLP levels are lower in the Δ*bamH* strain (Fig. 5a), while BamI and BamJ are not affected (Supplementary Table 4). In total, 14 OM β-barrels had lower abundance in the Δ*bamH* strain, 9 of which are TBDTs; 35 lipoproteins were downregulated in the Δ*bamH* strain, 24 of which are likely surface exposed based on the presence of a lipoprotein export signal.

The proteins with altered abundance in the *bamH* deletion strain relative to the wild type represent relatively small fractions of the total number of β-barrels and (S)LPs. The proteomics data therefore show that the BtBAM complex missing all its SLP components does not result in global depletion of OM β-barrels and SLPs, and it does not drastically affect the normal BtBAM β-barrel insertase function in vitro. To check for potential suppressor mutations that would mask the effects of *bamH* deletion, we performed next-generation

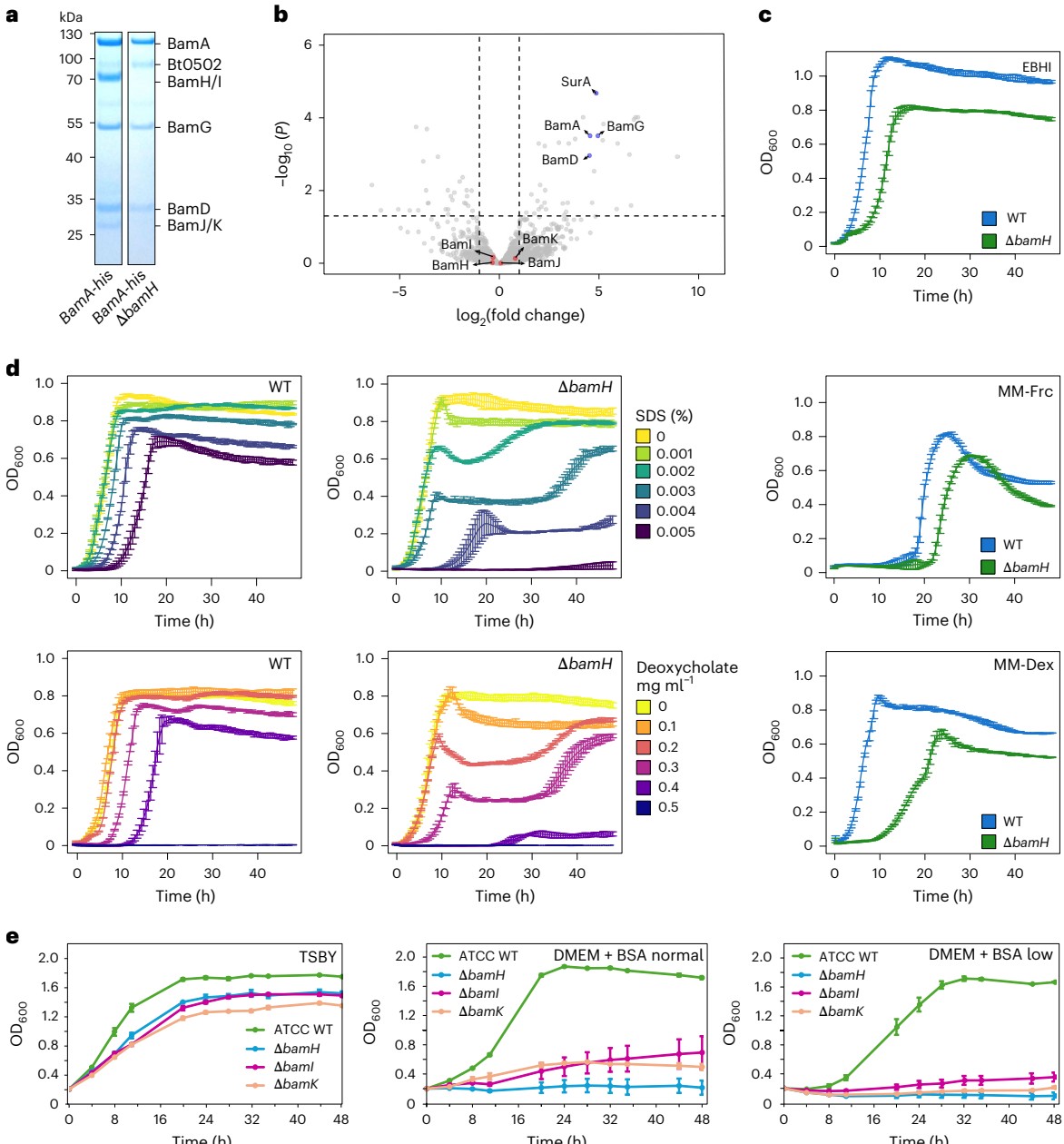

**Fig. 4 | Newly described BAM components are important for cell envelope integrity and growth in vitro. a**, Coomassie-stained 4–12% SDS-PAGE gel of BtBamA$_{his}$ pulldowns from wild-type cells and the *bamH* deletion strain. **b**, Volcano plot showing enrichment of proteins in pulldowns from *B. thetaiotaomicron bamA$_{his}$* Δ*bamH* versus the wild-type strain (*n* = 3 biological replicates for both conditions). The vertical dashed lines represent log$_2$(fold change) = ±1. Proteins above the horizontal dashed line have an adjusted *P* < 0.05. Statistical significance was inferred by two-sided *t*-test with Benjamini–Hochberg correction for multiple comparisons. **c**, Growth of wild-type and Δ*bamH B. thetaiotaomicron* strains in EBHI (top), minimal medium with 0.5% fructose (MM-Frc; middle) and minimal medium with 0.4% dextran 40 and 10%

enriched brain–heart infusion (MM-Dex; bottom). Each trace is an average of *n* = 3 technical repeats; the error bars represent the s.d. **d**, Growth of wild-type and Δ*bamH B. thetaiotaomicron* strains in EBHI in the presence of indicated concentrations of SDS and sodium deoxycholate. Each trace is an average of *n* = 3 technical repeats; the error bars represent the s.d. Each experiment is representative of 2 or 3 biological replicates. **e**, Growth of *P. gingivalis* wild type, Δ*bamH*, Δ*bamI* and Δ*bamJ* in rich medium (tryptic soy broth/yeast extract, TSBY) and minimal medium (DMEM) supplemented with BSA and regular or low amounts of vitamin K (0.5 mg l$^{-1}$ and 0.05 mg ml$^{-1}$, respectively) and haemin (5 mg ml$^{-1}$ and 0.5 mg ml$^{-1}$, respectively). Each data point is the average of *n* = 3 biological repeats; the error bars represent the s.d.

sequencing on *bamA-his* and *bamA-his* Δ*bamH* strains and found them to be identical, save for the absence of *bamH* in the latter. To test whether SLPs reach the cell surface in the Δ*bamH* strain, we used a proteinase K shaving assay on the abundant surface glycan binding lipoprotein (SGBP$^{lev}$, Bt1761 (ref. 46)) in wild-type and Δ*bamH* cells, and we found that SGBP$^{lev}$ is still surface exposed in the deletion strain (Fig. 5b). This suggests that the extracellular BtBAM dome is not required for SLP flipping.

The Bt1926-27 locus, which encodes a 100-kDa phase-variable SLP S-layer protein (Bt1927)[47,48], is much less abundant in the *B. thetaiotaomicron* Δ*bamH* strain. Its putative OM anchor or secretion partner (Bt1926)[47], predicted to form an 8-stranded β-barrel, was not detected in the Δ*bamH* strain and is thus absent from the volcano plot in Fig. 5a. Transmission electron microscopy images of thin sections of wild-type *B. thetaiotaomicron* show that the cells are surrounded by a diffuse density of variable width, which probably corresponds to the capsule,

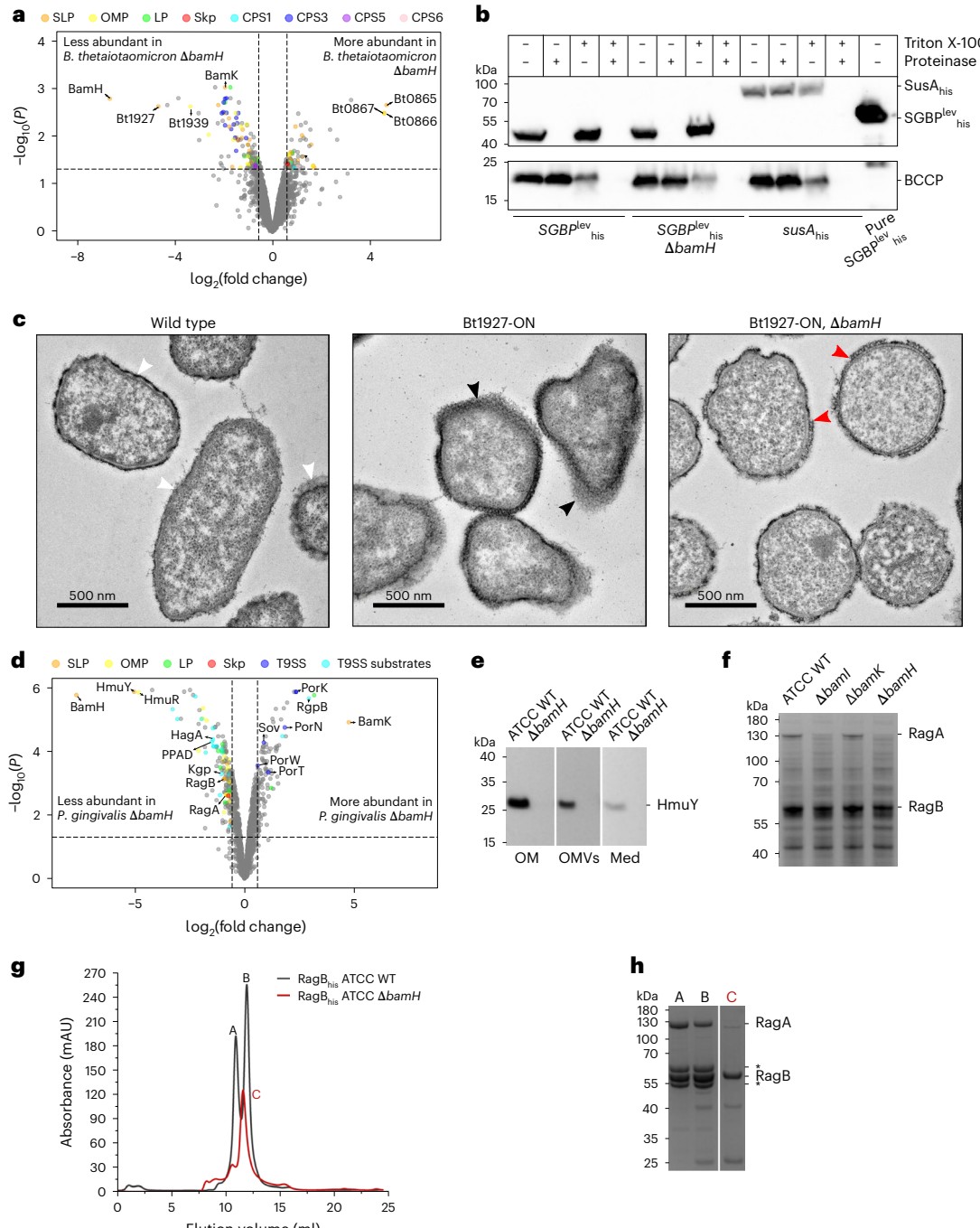

**Fig. 5 | Loss of extracellular BAM components results in pleiotropic changes at the cell surface. a**, Volcano plot showing wild-type *B. thetaiotaomicron* versus Δ*bamH* quantitative proteomics results (*n* = 3 biological replicates for each). The vertical dashed lines represent log$_2$(fold change) = ±0.58. Proteins above the horizontal dashed lines have an adjusted *P* < 0.05. Statistical significance was inferred by two-sided *t*-test with Benjamini–Hochberg correction for multiple comparisons. Data points of interest are labelled and colour coded according to function or cellular localization. **b**, *B. thetaiotaomicron* proteinase K shaving assay for testing protein surface exposure. The top immunoblot was probed with anti-His-HRP conjugate antibodies. The bottom panel shows the same immunoblot re-probed with StrepTactin-HRP. Biotin-carboxyl carrier protein (BCCP, Bt1688) is a cytoplasmic biotinylated protein. SusA is a periplasmic neopullulanase that was used as an OM lysis control. The experiment was done twice with similar results. **c**, Representative transmission electron microscopy images of thin-sectioned *B. thetaiotaomicron* cells (wild type *n* = 17, Bt1927-ON *n* = 15, Bt1927-ON, Δ*bamH n* = 26). The white arrowheads point to the capsule that varies in thickness between cells. The black arrowheads point to the S-layer

in Bt1927-ON cells. The red arrowheads point to the discontinuous S-layer in the Bt1927-ON, Δ*bamH* strain. **d**, Volcano plot showing wild-type *P. gingivalis* versus Δ*bamH* quantitative proteomics results (*n* = 3 biological replicates for each). Statistical significance was inferred by two-sided *t*-test with Benjamini–Hochberg correction for multiple comparisons. Changes in the T9SS and its substrates are described in Supplementary Discussion. **e**, Anti-HmuY immunoblot of OM, OMVs and culture medium (Med) fractions of wild type and Δ*bamH P. gingivalis* strains. The blot shown is representative of two biological repeats. **f**, SDS-PAGE analysis of *P. gingivalis* wild type, Δ*bamI*, Δ*bamK* and Δ*bamH* strain whole-cell lysates. **g**, RagB$_{his}$ pulldowns from *P. gingivalis* wild type and Δ*bamH* strains analysed by size-exclusion chromatography on a Superdex 200 Increase 10/300 GL column. The chromatograms shown are representative of two biological replicates. **h**, SDS-PAGE analysis of indicated peaks from **g**. The asterisks denote bands from peaks A and B that correspond to singly cleaved RagA, confirmed by mass spectrometry. SLP, surface-exposed lipoprotein; OMP, outer membrane protein; LP, lipoprotein; CPS, capsular polysaccharide; T9SS, type 9 secretion system; PPAD, peptidylarginine deiminase.

but there is no density attributable to an S-layer (Fig. 5c). This is because most wild-type *B. thetaiotaomicron* cells do not transcribe *bt1927* as the upstream invertible promoter is inactive, but the promoter can be engineered to make all cells produce the S-layer[47]. Micrographs of the engineered strain producing Bt1927 (Bt1927-ON) clearly show cells surrounded by an electron-dense, lattice-forming structure that corresponds to the S-layer (Fig. 5c and ref. 47). When *bamH* was deleted in Bt1927-ON, *B. thetaiotaomicron* formed only a partial, discontinuous S-layer, observed as patchy, electron-dense structures in micrographs (Fig. 5c). Thus, the extracellular dome of BtBAM has a role in the biogenesis of at least some cell surface structures involving β-barrels and SLPs.

### Deletion of *bamH* disrupts the main SLP–barrel complexes of *P. gingivalis*

As the genome of *P. gingivalis* encodes a relatively low number of proteins, we used whole cells instead of membrane fractions for quantitative proteomics. A total of 1,500 proteins were identified, providing excellent proteome coverage of 71%. The level of 146 proteins was significantly decreased ($\log_2$(fold change) < −0.58) (Fig. 5d, Supplementary Table 5 and Supplementary Discussion). Among the proteins with significantly reduced levels in the Δ*bamH* strain are 17 lipoproteins, 9 TBDTs and 9 other β-barrels (Fig. 5d and Supplementary Table 5). BamH, as expected, was the most depleted protein. The levels of other components of PgBAM remained unchanged, except for BamK, which intriguingly was the most enriched protein in the analysis. This suggests that the levels of the PgBAM SLPs are not decreased in the *bamH* mutant, even though they are probably not associated with the BamADG core.

There are two experimentally confirmed SLP–barrel complexes in *P. gingivalis*: the peptide transporter RagAB[44] and the haem uptake system HmuYR[49]. The levels of both the HmuY SLP and the HmuR TBDT are severely decreased in the Δ*bamH* strain (Fig. 5d and Supplementary Table 5). To verify the proteomics data, we probed western blots with anti-HmuY SLP antibodies and observed that HmuY was undetectable in the OM, outer membrane vesicles (OMVs) and culture medium in Δ*bamH* (Fig. 5e). The levels of the RagA OM β-barrel and the RagB SLP were moderately decreased to a similar extent in proteomics, even though RagA was barely visible on SDS–PAGE of a Δ*bamH* lysate compared with RagB (Fig. 5f). To check whether the RagAB complex is assembled in the Δ*bamH* strain, we isolated His-tagged RagB from a Δ*bamH* background (Fig. 5g). The SDS–PAGE profile revealed a clear RagB band, but almost no full-length RagA (Fig. 5h). Combined with a previous observation that the level of RagA is also much lower in a Δ*ragB* mutant[44], this suggests that RagA undergoes proteolysis due to the impaired assembly of the RagAB complex.

### Discussion

Many Pseudomonadota lack one or more accessory components of the *E. coli* BAM complex (BamB, BamC and BamE) and may therefore have other components[18]. However, in bacteria periplasm-facing BamF is the only non-canonical BAM-associated lipoprotein that has been identified to date and is present exclusively in Alphaproteobacteria[28]. In eukaryotes, mitochondria contain the analogous sorting and assembly complex, which in yeasts consists of the BamA orthologue Sam50 and accessory subunits Sam35 and Sam37. Sam35 and Sam37 are located in the cytosol (equivalent to the extracellular side in bacteria), interact with cytoplasmic loops of Sam50, and are crucial for the structural and functional integrity of the complex[50–52]. Plants and green algae contain the BamA orthologue Toc75, which assembles into a TOC–TIC supercomplex that spans both chloroplast membranes and bears very little resemblance to Pseudomonadotal BAM[53,54].

Here we have discovered that the BAM machinery in the Bacteroidota phylum has a previously undescribed architecture (Extended Data Fig. 10a), with an extended, essential core consisting of BamA, BamD and the BamA-associated β-barrel BamG. BamG serves as an attachment platform for an extracellular dome-like structure composed of four SLPs, BamH–K, none of which are essential in *B. thetaiotaomicron* and *P. gingivalis* in vitro. None of the newly described components resemble sorting and assembly and TOC–TIC components. Given that the defining difference between OMPs of Bacteroidota and Pseudomonadota is the dominance of SLPs in the former, we propose that the previously undescribed components and architecture of Bacteroidetal BAM may have evolved to enable the efficient biogenesis of SLP–barrel complexes (Extended Data Fig. 10b).

How the ubiquitous SLPs of Bacteroidota reach the cell surface is an important, unresolved question. In Pseudomonadota, there are several systems for which a function in secretion and surface exposure of lipoproteins has been proposed, such as the Slam[55], the type 2 secretion system in *Klebsiella* spp.[56], the Aat system for secretion of unique, glycine-acylated lipoproteins[57] and the OM LPS assembly protein LptD in *Borrelia burgdorferi*[58]. With the exception of LptD, *B. thetaiotaomicron* and some other Bacteroidota lack all of the aforementioned systems while its genome encodes >400 SLPs. Given the central roles of SLPs in Bacteroidota OM biology, it is likely that the SLP flippase is essential, although it is possible that multiple, non-essential flippases exist for subsets of SLPs. One possibility is that BamA is the Bacteroidota SLP flippase. There is precedent for such a role, as the stress sensor lipoprotein RcsF in *E. coli* is exported to the cell surface by EcBamA[59,60]. Identification of the SLP flippase(s) will remain an important area for further study.

The structure and positioning of the extracellular dome formed by BamH–K, and the involvement of two PPIs (BamJK), suggest that this structure may form an extracellular chaperone or assembly cage for the formation of SLP–barrel complexes that are a hallmark of the Bacteroidota phylum. The removal of this cage in Δ*bamH* kept the essential core BamADG complex intact but led to substantial growth and OM defects and changes in OMP abundance in both *B. thetaiotaomicron* and *P. gingivalis*. *B. thetaiotaomicron* lacking the dome is defective in S-layer production, possibly due to impaired attachment of the S-layer SLP Bt1927 to its putative anchor OM β-barrel Bt1926. Loss of BamH in *P. gingivalis* led to low levels of haem and peptide uptake systems, which explains the dramatic growth defects observed in minimal medium with BSA used as the carbon source. While the observed proteome changes may seem modest for *B. thetaiotaomicron* and more extensive in *P. gingivalis*, it should be noted that they result from removal of extracellular BAM components only. Moreover, the proteomics report on total protein abundance and do not provide clues about, for example, defective or slower assembly of SLP–barrel complexes. BamH is one of only ~80 core fitness components for mouse gut colonization by several *Bacteroides* spp.[61], suggesting that the effects of BamH removal would be more dramatic in vivo. Together, our functional data suggest that the extracellular dome has an important role in OMP complex and cell surface structure biogenesis in Bacteroidota through an unknown mechanism.

BAM has emerged as an important target for antibacterial therapy owing to its essential role in OMP biogenesis and maintaining bacterial cell integrity. Several types of BAM inhibitors have been developed, including small molecules, peptides, natural products and antibodies[62]. The mode of action of these compounds involves binding to critical regions of BamA (such as L3, L4, L6 loops and lateral seam) or interfering with the BamAD interface, thereby impairing proper assembly of the Pseudomonadota BAM complex and folding of OMPs[63,64,65]. To reach the periplasmic components of BAM, potential inhibitors must be able to cross the OM, which drastically limits the number of useful compounds[66]. Bacteroidota BAM is a promising target for inhibitors owing to the extracellular localization of the BamH–K dome, which makes it an easily accessible target. Moderate conservation of these surface-exposed components provides an opportunity to discover inhibitors targeting pathogenic Bacteroidota such as *P. gingivalis* while avoiding the commensals.

## Methods

### Bacterial strain culturing and genetic manipulation

Bacterial strains used in this study are listed in Supplementary Table 6. *E. coli* strains were grown at 37 °C in lysogeny broth (20 g l⁻¹, BioShop, LBL405) with 100 μg ml⁻¹ ampicillin (BioShop, AMP201) and on 1.5% agar plates. For constructing *B. thetaiotaomicron* chromosomal mutants and protein production, strains were cultured in enriched brain–heart infusion (EBHI), containing 37 g l⁻¹ brain–heart infusion powder (OXOIDLTD, CM1135B), 5 g l⁻¹ yeast extract and 1 μg ml⁻¹ haemin (Sigma-Aldrich, 51280), supplemented with 0.5 g l⁻¹ cysteine (Melford, C50010) and 1 μg ml⁻¹ vitamin K (Sigma-Aldrich, M5625). For growth curve and proteinase K experiments, defined minimal medium (MM) supplemented with 1 μg ml⁻¹ haemin and 0.5% carbon source was used, unless otherwise indicated. Bacteria were grown under anaerobic conditions at 37 °C in a Don Whitley A35 workstation. *B. thetaiotaomicron* genetic deletions and mutations were created by allelic exchange using the pExchange-tdk vector[67]. Briefly, the constructed pExchange-tdk plasmids, containing the mutations and deletions plus ~700 bp flanks up- and downstream (DNA inserts were synthesized by Twist Bioscience), were transformed into S17 λ pir *E. coli* cells, to achieve conjugation with the *B. thetaiotaomicron* recipient strain. The conjugation plates were scraped, and *B. thetaiotaomicron* cells that underwent a single recombination event were selected for by plating on BHI–haemin agar plates containing gentamicin (200 μg ml⁻¹, Melford, G38000) and erythromycin (25 μg ml⁻¹, Duchefa, E0122). Eight colonies were restreaked on fresh BHI–haemin–gentamicin–erythromycin plates. Single colonies were cultured in BHI–haemin overnight and pooled. To select for the second recombination event, pooled cultures were plated on BHI–haematin agar plates containing 5-fluorodeoxyuridine (FUdR; 200 μg ml⁻¹, Thermo Scientific, L16497.ME). After 48–72 h of growth, FUdR-resistant colonies were restreaked on fresh BHI–haematin–FUdR. From these, single colonies were cultured in BHI and genomic DNA was extracted and screened for the correct mutations using diagnostic PCR. This procedure was used to introduce the N-terminal His₇-tag on BtBamA, and *bamH* and *bamI* deletions into the chromosome of *B. thetaiotaomicron* VPI-5482 *tdk⁻* and BamA$_{his}$, as well as a *bamH* deletion in the Bt1927-ON background.

*P. gingivalis* strains were grown anaerobically (90% nitrogen, 5% carbon dioxide and 5% hydrogen) at 37 °C in enriched tryptic soy broth (eTSB; 30 g l⁻¹ tryptic soy broth (Sigma-Aldrich, T8907), 5 g l⁻¹ yeast extract (BioShop, YEX401), 5 mg l⁻¹ haemin (Sigma-Aldrich, H9039), 0.5 mg l⁻¹ menadione (Sigma-Aldrich, M5625) and 0.25 g l⁻¹ L-cysteine (BioShop, CYS342)) or on eTSB blood agar plates with 1.5% agar (Bio-Shop, AGR001) and 5% defibrinated sheep blood (Biomaxima SL0160). For mutant selection, appropriate antibiotics were added: erythromycin (Sigma-Aldrich, E6376) at 5 μg ml⁻¹ or tetracycline (BioShop, TET701) at 1 μg ml⁻¹. *P. gingivalis* mutants were generated through homologous recombination. For deletion mutants, 1-kb fragments flanking the *bamH*, *bamI* or *bamK* genes and chosen antibiotic resistance cassettes were amplified by PCR and cloned into pUC19 plasmid using restriction cloning. For expression of C-terminally-His-tagged BamH protein, master plasmid was first generated. A 1-kb fragment coding the C-terminal part of BamH and a 1-kb fragment downstream of it as well as an antibiotic resistance cassette were amplified and cloned into pUC19 plasmid using restriction cloning. The 7His-tag was then introduced into the master plasmid by PCR. Plasmids and primers used for mutant construction are listed in Supplementary Tables 7 and 8, respectively. Plasmids were electroporated into *P. gingivalis* competent cells, followed by plating the cells on eTSB blood agar with appropriate antibiotics and growth for 10 days. All generated plasmids were confirmed by PCR and Sanger sequencing. *B. thetaiotaomicron* and *P. gingivalis* clones were screened by PCR and verified by Sanger sequencing.

### BtBAM expression and purification

The *B. thetaiotaomicron* bamA$_{his}$ strain was cultured overnight in EBHI and used to inoculate 10 l of rich medium (25 g l⁻¹ brain–heart infusion powder, 3.5 g l⁻¹ yeast extract, 1 μg ml⁻¹ vitamin K, 0.5 g l⁻¹ L-cysteine) that was equilibrated at 37 °C overnight in an anaerobic chamber. For each 500-ml bottle of pre-warmed medium, 3 ml of overnight culture was used. The cultures were grown for 5–7 h until late exponential phase (optical density at 600 nm (OD₆₀₀) ~1.8–2.0). Cultures were pelleted by centrifugation at 6,000 g for 30 min at 4 °C, and the pellets were stored at −20 °C.

Cell pellets were processed in 2-l batches to maximize yield. Cells were thawed, homogenized in Tris-buffered saline (TBS; 20 mM Tris–HCl pH 8.0, 300 mM NaCl), supplemented with DNase I (Roche, 04716728001) and lysed by a single pass through a cell disruptor at 22–23 kpsi. The total membrane fraction was isolated by ultracentrifugation at 200,000 g for 50 min at 4 °C. Membranes were homogenized and solubilized in 30 ml TBS with 0.75% DDM (Anatrace, D310S) and 0.75% decyl maltoside (DM, Anatrace, D322S) for 1 h at 4 °C with stirring. Insoluble material was pelleted by ultracentrifugation at 200,000 g for 30 min at 4 °C. The supernatant was loaded on a 1.5-ml chelating Sepharose (Cytiva, 17057502) gravity flow column charged with Ni²⁺ ions at 4 °C and equilibrated with TBS supplemented with 0.15% DDM. The column was washed with 30 column volume (CV) TBS supplemented with 30 mM imidazole and 0.15% DDM, and bound material was eluted with 4 CV TBS supplemented with 200 mM imidazole and 0.15% DDM. The eluate was concentrated using an Amicon Ultra filtration device column (100 kDa molecular weight cut-off, Sigma-Aldrich, UFC9100) and loaded on a Superdex 200 Increase 10/300 GL (Cytiva, 28990944) equilibrated in 10 mM HEPES–NaOH, 100 mM NaCl and 0.03% DDM pH 7.5. Peak fractions were analysed by SDS-PAGE and Coomassie staining for purity. Gel bands of interest were cut out and identified by peptide fingerprinting at the Metabolomics and Proteomics Laboratory, University of York. Material purified from five separate 2-l batches was combined and subjected to a final size-exclusion chromatography run as above. The BtBAM complex was concentrated, flash-frozen in liquid nitrogen and stored at −80 °C.

### BtBAM cryo-EM structure determination

Purified BtBAM complexes at 8 mg ml⁻¹ were applied to glow-discharged Quantifoil R1.2/1.3 holey carbon 200 mesh Cu grids (Oxford Instruments, 51-1625-0131), followed by blotting and plunge-freezing in liquid ethane using a Vitrobot Mark IV device at 4 °C and 100% humidity. Data were collected on a Titan Krios electron microscope operating at 300 kV accelerating voltage on a Falcon 4i direct electron detector with a Selectris energy filter (10 eV slit). Videos were recorded in electron-event representation format at a magnification of 165,000, corresponding to a sampling rate of 0.74 Å per pixel. The defocus was varied between −0.8 μm and −2.0 μm during data collection. The total dose for each video was ~35 e⁻ Å⁻². A total of 13,558 videos were collected in total. Data collection parameters are summarized in Supplementary Table 2.

The cryo-EM data processing workflow is shown in Supplementary Fig. 1. All data processing was done in cryoSPARC v4.5.3 (ref. 68). Videos were patch motion corrected, followed by patch contrast transfer function (CTF) correction. Particles were picked manually to generate 2D classes for template-based picking. In total, 2,266,553 particles were extracted in 224 × 224 pixel boxes (1.48 Å per pixel) and subjected to two rounds of 2D classification. Particles from 2D classes resembling protein density were pooled into a single particle stack containing 308,834 images, which was then used for ab initio reconstruction with 4 classes. 2D classes containing 57,300 noise and junk particles were used for ab initio reconstruction of 4 decoy classes. The 308,834-particle stack was subjected to heterogeneous refinement using the 8 ab initio classes as initial volumes. A total of 137,389 particles from the best two classes from the first round of heterogeneous refinement were subjected to 3D variability analysis[32] and to another round of heterogeneous refinement against the same initial volumes as in the first round. This resulted in 2 classes with clear protein features: class 1 with 49,137 particles with

no periplasmic density and class 2 with 55,224 particles and periplasmic density. Particles belonging to these two classes were independently used in non-uniform refinement[69], yielding 3.41 Å (class 1) and 3.49 Å (class 2) reconstructions. Refined particles were re-extracted in 448 × 448 pixel boxes (0.74 Å per pixel) and subjected to another round of non-uniform refinement with enabled per-particle defocus and global CTF parameter (tilt and trefoil) refinement. A total of 48,843 particles from class 1 yielded a final reconstruction at 3.28 Å; 56,321 particles from class 2 yielded a final reconstruction at 3.46 Å. Global resolution was estimated using gold-standard Fourier shell correlation curves and the 0.143 criterion (Supplementary Fig. 1c). Particle viewing direction distribution plots and local resolution estimates are given in Supplementary Fig. 1d–f.

BtBamHIJK AF2 models were docked into the class 1 volume and subjected to cycles of manual adjustment in Coot[70] and real space refinement in Phenix[71]. Mannose monosaccharides were placed in O-glycan densities to prevent the protein model from moving into the glycan density. Similarly, BtBamADG AF2 models were docked, manually adjusted and refined against the class 2 volume. The refined BtBamHIJK class 1 model was rigid-body-refined against the class 2 volume, which had much poorer density for the BtBamHIJK region than class 1 (Supplementary Fig. 1f), to obtain the complete BtBAM model. A composite map incorporating the best parts of the volume representing each class was generated using phenix.combine_focused_maps. Model refinement statistics are shown in Supplementary Table 2.

### PgBAM cryo-EM structure determination

Purified PgBAM complexes at 4.9 mg ml$^{-1}$ were applied to glow-discharged Quantifoil R2/1 holey carbon Cu grids (200 mesh) (EM Resolutions, QR21200Cu100), followed by blotting and plunge-freezing in liquid ethane using a Vitrobot Mark IV (Thermo Fisher Scientific) set to 95% humidity and 4 °C with the following blotting parameters: 2 s blot time, 0 s wait time, blot force −4, 0 s drain time and a total blot of 1. Data were collected using a Titan Krios G3i (Thermo Fisher Scientific; Solaris) electron microscope operating at 300 kV accelerating voltage on a Gatan K3 Summit direct electron detector with a Gatan Quantum energy filter. Videos were recorded in TIFF format at a magnification of ×105,000, corresponding to a sampling rate of 0.84 Å per pixel. The defocus was varied between −0.6 μm and −1.5 μm during data collection. The total dose for each video was -41 e$^−$ Å$^{-2}$. In total, 11,459 videos were collected. Data collection parameters are summarized in Supplementary Table 2.

The cryo-EM data processing workflow is shown in Supplementary Fig. 3. All data processing was done in cryoSPARC v4.5.3. Videos were patch motion corrected, followed by patch CTF correction. Three rounds of 2D classification were performed to generate 2D classes for template-based picking. A total of 4,830,281 particles were extracted in 200 × 200 pixel boxes (1.68 Å per pixel) and subjected to two rounds of 2D classification. Particles from 2D classes resembling protein density were pooled into a single particle stack containing 349,717 images, which was then used for ab initio reconstruction with 4 classes. 2D classes containing 9,273 noise and junk particles were used for ab initio reconstruction of 3 decoy classes. Particles were re-extracted in 384 × 384 pixel boxes (0.84 Å per pixel). The 104,912-particle stack was subjected to heterogeneous refinement using the 5 ab initio classes as initial volumes. A total of 101,685 particles from the best 2 classes were combined and subjected to ab initio reconstruction (2 classes) with the class similarity parameter set to 0.0001. The class containing 86,584 particles was used in non-uniform refinement, yielding a 3.45-Å reconstruction. Refined particles were subjected to local CTF refinement and global CTF refinement and used again in non-uniform refinement, yielding a final reconstruction at 3.24 Å (PgBAM without periplasmic density). For PgBAM with periplasmic density, after the 2 initial rounds of 2D classification following template picking, the particles were subjected to a further 5 rounds of 2D classification

(Supplementary Fig. 3). The resulting 13,929 particles were subjected to heterogeneous refinement with one protein and 2 decoy volumes. Protein class was used in non-uniform refinement, yielding a final reconstruction at 4.26 Å (PgBAM with periplasmic density). Global resolution was estimated using gold-standard Fourier shell correlation curves and the 0.143 criterion (Supplementary Fig. 3c). Particle viewing direction distribution plots and local resolution estimates are shown in Supplementary Fig. 3d–f. AF2 models of all subunits were docked into the final volume of PgBAM without periplasmic density and subjected to cycles of manual adjustment in Coot and real space refinement in Phenix. Mannose monosaccharides were placed in O-glycan densities to prevent the protein model from moving into the glycan density. Model refinement statistics are shown in Supplementary Table 2.

### Growth curves

Overnight EBHI cultures of *B. thetaiotaomicron* were used to inoculate 5 ml of fresh EBHI (1:25), followed by incubation at 37 °C for 3–4 h in an anaerobic cabinet. The cells were collected by centrifugation for 3 min at 2,900 $g$, 20 °C, and resuspended in 1 ml pre-warmed MM. Glass tubes with EBHI or MM (including carbon source) and supplements indicated in the main text were inoculated with the resuspended cells at OD$_{600}$ = 0.04, and the cultures were dispensed in 0.2-ml aliquots in sterile 96-well clear polystyrene cell culture plates (Corning Costar, 3599). Medium without any added cells was included as blank. Plates were incubated anaerobically at 37 °C for 30 min and sealed with a Breathe-Easy gas-permeable seal (Diversified Biotech, BEM-1), followed by growth monitoring for 48 h in a Biotek Epoch microplate reader housed inside an anaerobic cabinet. Blank well readings were subtracted from wells with added cells, and triplicate readings were averaged. The experiments were repeated at least two times on different days to ensure consistency.

*P. gingivalis* Δ*bamH*, Δ*bamI* and Δ*bamK* mutants were first grown in eTSB overnight. The cultures were resuspended in fresh eTSB or washed and resuspended in DMEM (Thermo Fisher Scientific, A1443001) supplemented with 1% BSA (BioShop, ALB001), 0.25 g l$^{-1}$ L-cysteine and either regular or decreased amounts of vitamin K (0.5 mg l$^{-1}$ and 0.05 mg ml$^{-1}$, respectively) and haemin (5 mg l$^{-1}$ and 0.5 mg l$^{-1}$, respectively). In each case, the OD$_{600}$ was adjusted to 0.2 and the growth was monitored for 48 h through OD$_{600}$ measurements at predetermined time intervals, as *P. gingivalis* does not grow in multi-well plates.

### PgBAM expression and purification

The PgBAM complex was isolated from His-tagged BamH-expressing *P. gingivalis* according to a previously described protocol[44]. Briefly, cells from 5 l were collected by centrifugation at 6,500 $g$ for 40 min and resuspended in TBS with 1 mM $N_\alpha$-tosyl-L-lysine chloromethyl ketone (TLCK, Sigma-Aldrich, T8907), 1× cOmplete EDTA-free Protease Inhibitor Cocktail (Roche, 11873580001) and 20 μg ml$^{-1}$ DNase I (Roche, 11284932001), followed by lysis with Constant Systems Cell Disruptor (Model TS2) at 23 kpsi. The total membrane fraction was separated from the soluble fraction through ultracentrifugation at 200,000 $g$ for 1 h and then homogenized and solubilized in TBS with 1% n-dodecyl β-D-maltoside (DDM, Anatrace, D310) at 4 °C overnight. The extract was clarified by ultracentrifugation at 200,000 $g$ for 50 min and loaded on 1.5 ml pre-equilibrated nickel-affinity resin (Chelating Sepharose, GE Healthcare, 17531802). The column was washed with 15 CV of TBS containing 0.2% DDM and 30 mM imidazole (BioShop, IMD508), and the bound protein was eluted with 2 CV of TBS containing 0.2% DDM and 250 mM imidazole. The complex was further purified by size-exclusion chromatography in 10 mM HEPES pH 8.0, 100 mM NaCl and 0.03% DDM using a Superdex 200 Increase 10/300 GL column (Sigma-Aldrich, GE28-9909-44). In pulldown experiments shown in Fig. 5g,h, RagB-His from wild-type and Δ*bamH* cells was purified in the same manner, except lauryldimethylamine-N-oxide (Anatrace, D360) was used instead of DDM.

## Pulldown proteomics sample preparation, mass spectrometry and analysis

Protein solutions were diluted 1:1 with aqueous 10% (v/v) SDS and 100 mM triethylammonium bicarbonate (TEAB, Sigma-Aldrich, T7408). Protein was reduced with 5.7 mM tris(2-carboxyethyl)phosphine (TCEP, Thermo Fisher Scientific, T2556) and heating to 55 °C for 15 min before alkylation with 22.7 mM methyl methanethiosulfonate (Sigma-Aldrich, 208795) at room temperature for 10 min. Protein was acidified with 6.5 µl of aqueous 27.5% (v/v) phosphoric acid then precipitated with dilution sevenfold into 100 mM TEAB 90% (v/v) methanol. Precipitated protein was captured on an S-trap (Protifi–C02-micro) and washed five times with 165 µl 100 mM TEAB 90% (v/v) methanol before digesting with the addition of 20 µl 0.1 µg µl$^{-1}$ Promega Trypsin/Lys-C mix (V5071) in aqueous 50 mM TEAB and incubation at 47 °C on a hot plate for 2 h. Peptides were recovered from the S-trap by spinning at 4,000 $g$ for 60 s. S-traps were washed with 40 µl aqueous 0.2% (v/v) formic acid and 40 µl 50% (v/v) acetonitrile:water and the washes were combined with the first peptide elution. Peptide solutions were dried in a vacuum concentrator then resuspended in 20 µl aqueous 0.1% (v/v) formic acid. Peptides were loaded onto EvoTip Pure tips for desalting and as a disposable trap column for nanoUPLC using an EvoSep One system. A pre-set EvoSep 100 SPD gradient (from Evosep One HyStar Driver 2.3.57.0) was used with an 8-cm EvoSep C$_{18}$ Performance column (8 cm × 150 µm × 1.5 µm). The nanoUPLC system was interfaced to a timsTOF HT mass spectrometer (Bruker) with a CaptiveSpray ionization source (Source). Positive parallel accumulation serial fragmentation-data-dependent acquisition (PASEF-DDA), electrospray ionisation (ESI)-mass spectrometry (MS) and MS$^2$ spectra were acquired using Compass HyStar software (version 6.2, Bruker). Instrument source settings were as follows: capillary voltage, 1,500 V; dry gas, 3 l min$^{-1}$; and dry temperature, 180 °C. Spectra were acquired between $m/z$ 100 and 1,700. Trapped ion mobility spectrometry settings were as follows: inverse reduced ion mobility rate (1/$K_0$), 0.6–1.60 V s cm$^{-2}$; ramp time, 100 ms; and ramp rate, 9.42 Hz. Data-dependant acquisition was performed with 10 PASEF ramps and a total cycle time of 1.17 s. An intensity threshold of 2,500 and a target intensity of 20,000 were set with active exclusion applied for 0.4 min after precursor selection. Collision energy was interpolated between 20 eV at 0.6 V s cm$^{-2}$ and 59 eV at 1.6 V s cm$^{-2}$.

Liquid chromatography (LC)-MS data, in Bruker.d format, were processed using DIA-NN (1.8.2.27) software and searched against an in silico predicted spectral library, derived from the *B. thetaiotaomicron* subset of UniProt (UP000001414, 2024/06/10) appended with common proteomic contaminants. Search criteria were set to maintain a false discovery rate of 1%. High-precision quant-UMS[72] was used for extraction of quantitative values within DIA-NN. Peptide-centric output in .tsv format, was pivoted to protein-centric summaries using KNIME 5.1.2 and data filtered to require protein $q$-values < 0.01 and a minimum of two peptides per accepted protein. Calculation of log$_2$ fold difference and differential abundance testing was performed using limma via FragPipe-Analyst[73]. Sample minimum imputation was applied and the Hochberg and Benjamini approach was used for multiple test correction. Volcano plots were made using the EnhancedVolcano[74] package in R.

## Quantitative proteomics sample preparation, mass spectrometry, and data analysis

For *B. thetaiotaomicron* membrane proteomics, three independent 0.5 l cultures of wild type and Δ*bamH* strains were grown and processed up to the membrane isolation step as described above for BtBAM complex purification. For *P. gingivalis* whole-cell proteomics, three independent wild-type and Δ*bamH P. gingivalis* cultures were grown overnight in eTSB. Cells from 50 ml of culture were washed three times with ice-cold PBS. Aliquots corresponding to 50 ml culture of OD$_{600}$ = 0.6 were centrifuged, and the pellets were stored at −80 °C until needed.

*P. gingivalis* and *B. thetaiotaomicron* cell and membrane pellets were lysed by addition of 50 mM TEAB pH 8.5 with 5% (w/v) SDS and sonication using a UP200St ultrasonic processor (Hielscher) at 90 W, 45 s pulse and 15 s rest three times. *B. thetaiotaomicron* lysis buffer included 1× cOmplete protease inhibitor cocktail (Roche, 11836170001), for *P. gingivalis*, 2× protease inhibitor cocktail and 100 mM N-ethylmaleimide (NEM, Sigma-Aldrich, 34115-M). Samples were then denatured with 5 mM TCEP at 60 °C for 15 min, alkylated with 30 mM iodoacetamide (*B. thetaiotaomicron*) or 10 mM NEM (*P. gingivalis*) at room temperature for 30 min in the dark, and acidified to a final concentration of 2.7% phosphoric acid. Samples were then diluted eightfold with 90% MeOH 10% TEAB (pH 7.2) and added to the S-trap micro columns. The manufacturer-provided protocol was then followed, with a total of five washes in 90% MeOH 10% TEAB (pH 7.2) and trypsin added at a ratio of 1:10 enzyme:protein and digestion performed for 18 h at 37 °C. Peptides were dried using a vacuum concentrator and stored at −80 °C, and, immediately before mass spectrometry, were resuspended in 0.1% formic acid.

LC was performed using an Evosep One system with a 15-cm Aurora Elite C18 column with integrated captivespray emitter (IonOpticks), at 50 °C. Buffer A was 0.1% formic acid in HPLC water; buffer B was 0.1% formic acid in acetonitrile. Immediately before LC–MS, peptides were resuspended in buffer A and a volume of peptides equivalent to 500 ng was loaded onto the LC system-specific C18 EvoTips, according to manufacturer instructions, and subjected to the predefined Whisper100 20 SPD protocol (in which the gradient is 0–35% buffer B, 100 nl min$^{-1}$, for 58 min; 20 samples per day are permitted using this method). The Evosep One was used in line with a timsToF-HT mass spectrometer (Bruker), operated in diaPASEF mode. Mass and IM ranges were 300–1,200 $m/z$ and 0.6–1.4 1/$K_0$; diaPASEF was performed using variable width IM-$m/z$ windows, as described previously[75]. TIMS ramp and accumulation times were 100 ms; total cycle time was ~1.8 s. Collision energy was applied in a linear fashion, where ion mobility = 0.6–1.6 1/$K_0$ and collision energy = 20–59 eV.

Raw diaPASEF data files were searched using DIA-NN V 1.8.2 beta 27 (ref. [76]), using its in silico-generated spectral library function, based on reference proteome FASTA files for *B. thetaiotaomicron* (UP000001414, downloaded from UniProt on 1 December 2021) or *P. gingivalis* (UP000008842, downloaded from UniProt on 13 February 2024) and a common contaminants list[77]. Trypsin specificity with a maximum of 2 variable modifications and 1 missed cleavage were permitted per peptide, cysteine carbamidomethylation (*B. thetaiotaomicron*) or NEM (UniMod:108, *P. gingivalis*) was set as fixed, and oxidation of methionine was set as a variable modification. Peptide length and $m/z$ were 7–30 and 300–1,200; charge states 2–4 were included. Mass accuracy was fixed to 15 ppm for MS1 and MS2. Protein and peptide false discovery rates were both set to 1%. Match between runs and real-time normalization were used for *B. thetaiotaomicron* but not for *P. gingivalis*. All other settings were left as defaults. The protein group matrix outputs were processed (separately for each species) in R using Rstudio, with filtering to exclude contaminants, and include a minimum of 2 peptides per protein group, and proteins present in a minimum of 2 out of 3 replicates in any one condition. Statistical analysis was performed using Limma[78], with statistical significance inferred by $t$-test with Benjamini–Hochberg correction for multiple comparisons. Volcano plots were made using the EnhancedVolcano[74] package in R.

## *B. thetaiotaomicron* proteinase K shaving assay

Overnight EBHI cultures were used to inoculate fresh minimal medium supplemented with 0.5% fructose, or 0.5% maltose for the control *susA*$_{his}$ strain, at OD$_{600}$ ~0.1. After a 4-h incubation at 37 °C in an anaerobic chamber, 33 OD units of each strain were collected by centrifugation for 5 min at 4,000 $g$. Cell pellets were resuspended in 1 ml PK buffer (50 mM Tris–HCl pH 8.0, 50 mM NaCl, 1 mM MgCl$_2$), split into four 250-µl aliquots and

pelleted again. Two aliquots were resuspended in 180 μl PK buffer and supplemented with 20 μl PK buffer or 20 mg ml⁻¹ fresh proteinase K solution (Sigma-Aldrich, P2308). The other two aliquots were resuspended in 180 μl PK buffer containing 1% Triton X-100 (v/v) (Fisher BioReagents, BP151-100) and supplemented with 20 μl PK buffer or 20 mg ml⁻¹ fresh proteinase K solution. Proteinase K digestion was carried out at 37 °C for 2 h under aerobic conditions. Samples containing Triton X-100 were supplemented with 5 mM phenylmethylsulfonyl fluoride (PMSF) and incubated at 20 °C for 15 min. Samples without detergent were pelleted, washed twice with PK buffer containing 2 mM PMSF and lysed in 200 μl PK buffer with 1% Triton X-100 and 2 mM PMSF for 10 min at 20 °C. Samples were mixed with SDS loading buffer, boiled immediately for 10 min and separated on a 12% FastCast gel (Bio-Rad). Proteins were transferred from the gel to a 0.2-μm polyvinylidene fluoride membrane using a Bio-Rad Trans-Blot Turbo system. The membrane was stained with Ponceau S to confirm successful transfer, blocked for 1 h with 3% BSA solution in PBST (phosphate buffered saline with 0.1% Tween-20) and incubated for 2 h with anti-His-HRP conjugate monoclonal antibody (Roche, 11965085001) diluted 1:500 in blocking solution. The membrane was washed three times with PBST and HRP was detected using chemiluminescence. The membrane was then re-probed with StrepTactin-HRP (Bio-Rad) in blocking solution (1:10,000) for 2 h, and washed and developed as above.

### *P. gingivalis* cell fractionation

*P. gingivalis* strains were grown overnight in eTSB. All cultures were adjusted to the same OD₆₀₀ and centrifuged at 7,500 g for 40 min. The supernatant was further ultracentrifuged at 200,000 g for 1 h to separate OMVs from media. OMV-free media were concentrated 20×. OMVs were resuspended in PBS with 1 mM TLCK and sonicated. Cells were washed and resuspended in PBS with 1 mM TLCK, followed by sonication and centrifugation at 7,500 g for 15 min. Supernatant samples were collected and processed as for the soluble fraction. Pellets, corresponding to the insoluble fraction, were washed and resuspended in PBS with 1 mM TLCK.

### SDS–PAGE sample preparation and western blotting analysis

Samples of *P. gingivalis* OMVs, OMV-free media, and soluble and insoluble fractions were mixed with NuPAGE LDS Sample Buffer (Thermo Fisher Scientific, NP0007) and boiled at 95 °C for 5 min. Dithiothreitol (BioSHop, DTT001) was added to 50 mM, and the samples were boiled again at 95 °C for 5 min. Samples were separated in NuPAGE Bis-Tris Mini Protein Gels, 4–12% (Thermo Fisher Scientific, NP0323BOX) and either stained with Coomassie Brilliant Blue G250 (BioShop, CBB555) or electrotransferred onto nitrocellulose membranes (Sigma-Aldrich, GE10600001) in 25 mM Tris, 192 mM glycine and 20% methanol at 100 V for 60 min. To visualize total protein, membranes were stained with 0.1% Ponceau S (BioShop, PON001) in 1% acetic acid and rinsed with distilled water. Membranes were blocked with 5% skim milk (SM, BioShop, SKI400) in TBS with 0.1% Tween-20 (TTBS) at 4 °C overnight followed by 1 h incubation with anti-HmuY antibodies (1 μg ml⁻¹ in TTBS) and 50 min incubation with secondary HRP-conjugated anti-rabbit antibodies (Sigma-Aldrich, A0545) diluted 1:20,000 at room temperature. Signal detection was carried out with Pierce ECL western blotting substrate (Thermo Fisher Scientific, 32109).

### Transmission electron microscopy with sectioned *B. thetaiotaomicron* cells

Cells were cultured to stationary phase (OD₆₀₀ = 1.5–1.8), collected by centrifugation (5,000 × g, 5 min) and washed twice with 1 mM HEPES pH 7.0, 55 mM sucrose (BioShop, SUC700). Samples were fixed in 1.5% (v/v) glutaraldehyde (Sigma-Aldrich, G6257), 0.5% OsO₄ (Sigma-Aldrich, 208868), 0.15% ruthenium red (Sigma-Aldrich, 557450) and 55 mM sucrose in 1 mM HEPES pH 7.0. Next, samples were dehydrated in an ethanol series (50%, 70%, 90%, 96%, 100%, 2 × 15 min each) and in

propylene oxide (2 × 10 min) before embedding in a mixture of epoxy resins (Poly/bed 812, MNA, DDSA) (Polysciences, 08791). The following day, the preparations were transferred to DMP-30 resin (Polysciences, 00553) and polymerized in embedding forms at 60 °C. Embedded samples were sectioned with an EM UC7 microtome (Leica). Sections were contrasted on the grids using uranyl acetate and lead citrate. A Tecnai Osiris electron microscope (FEI) operating at an accelerating voltage of 200 kV was used to image the stained sections. Gatan Microscopy Suite (GMS) software v3.4.3 was used to take images on a Rio16 camera (Gatan).

### Bioinformatics

To determine the taxonomic distribution of newly discovered BAM components, protein sequences were first subjected to SSDB Motif Search in the Kyoto Encyclopedia of Genes and Genomes database. Then, the conserved motif was used as a query in the Enzyme Function Initiative–Enzyme Similarity Tool[79]. Default parameters and the UniProt database were used for searches.

Homologues to BtBamG (Bt4367) were identified through complementary BlastP searches performed at the NCBI Blast server against the non-redundant protein sequences (nr)[80]. BtBamG and selected BamG homologues were used as queries in BlastP unrestricted and taxonomically restricted searches to identify a representative set of BamG homologues across sequence and taxonomic diversity. A selection of BamG homologues were combined to generate protein alignments, optimizing taxonomic and sequence diversity representation for phylogenetic inferences. SeaView v5.0.5 (ref. 81) was used to infer, manipulate and visualize the alignments to generate figures and initial phylogenies. Clustal Omega was used to generate the protein alignments[82], and Gblocks[83] was used to trim the alignments to select the sites with the strongest hypothesis of homology and to remove excessively divergent regions before phylogenetic inference. The maximum likelihood framework for protein phylogenetics implemented in IQ-TREE was used for tree inferences[84,85]. Support values for branches were estimated using ultrafast bootstrapping[86]. The best-fitting single-matrix homogenous substitution model for the alignment was inferred with the automatic model selection (function 'Auto'), which was LG + G4 + I + F (Optimal pinv = 0.045, alpha = 1.879) for the Akaike information criterion (AIC) and corrected AIC and the Bayesian information criterion.

To investigate the distribution of BamG homologues across Bacteroidota genomes, the annotated proteins from complete genomes from the RefSeq genome database at the NCBI[87] were downloaded (1,448 genomes on 17 September 2024). A local BLASTP 2.12.0+ search[88,89] was performed using BtBamG as query (*B. thetaiotaomicron* VPI-5482, accession AAO79472.1 putative outer membrane protein, length = 431 residues). Blast hits with *E*-value ≤ 0.05 were recorded and are listed in Supplementary Table 2. tBlastn searches using the NCBI Blast portal were performed against individual genomes without strong BlastP hits (Supplementary Discussion).

### Reporting summary

Further information on research design is available in the Nature Portfolio Reporting Summary linked to this article.

## Data availability

Pulldown proteomic mass spectrometry datasets and result files are referenced in ProteomeXchange (PXD058163) and available to download from MassIVE (MSV000096498; https://doi.org/10.25345/C57W67J00). *B. thetaiotaomicron* membrane and *P. gingivalis* whole-cell mass spectrometry proteomics data have been deposited to the ProteomeXchange Consortium via the PRIDE[90] partner repository with the dataset identifiers PXD058903 for *B. thetaiotaomicron* and PXD058905 for *P. ginigivalis*. Cryo-EM reconstructions and atomic coordinate files with the indicated accession numbers have been uploaded to the Electron Microscopy Data Bank (EMDB) and the

Protein Data Bank (PDB): BtBAM class 1 (EMD-52200 and 9HIS), BtBAM class 2 (EMD-52202 and 9HIV), BtBAM composite map and model (EMD-52209 and 9HJ3), PgBAM class 1 (EMD-52218 and 9HJM) and PgBAM class 2 (EMD-52219). All data that support the findings of this study are available within the Article and Supplementary Information. Source data are provided with this paper.

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

## Acknowledgements

We thank M. Fischbach (Stanford University) for providing the Bt1927-ON strain and J. Abellon-Ruiz (Newcastle University) for providing the *B. thetaiotaomicron* SusA$_{his}$ strain. B.v.d.B. was financially supported by a Wellcome Trust Investigator award (214222/Z/18/Z), providing salary support to A.S. The York Centre of Excellence in Mass Spectrometry was created thanks to a major capital investment through Science City York, supported by Yorkshire Forward with funds from the Northern Way Initiative, and subsequent support from the Engineering and Physical Sciences Research Council (EP/K039660/1 and EP/M028127/1). We acknowledge the use of cryo-EM facilities at the Astbury Biostructure Laboratory, which was financially supported by the University of Leeds and the Wellcome Trust (108466/Z/15/Z and 221524/Z/20/Z), and thank Y. Halfon for support. This study was supported by a Polish National Science Centre grant to M.M. (UMO-2023/51/D/NZ1/02675). M.M. was also supported by the Bekker programme of the Polish National Agency for Academic Exchange (PPN/BEK/2020/1/00103/U/00001). We acknowledge the Polish high-performance computing infrastructure PLGrid (HPC Center: ACK Cyfronet AGH) for access to computational facilities and support (PLG/2023/016774). This publication was partially developed under the provision of the Polish Ministry and Higher Education project 'Support for research and development with the use of research infrastructure of the National Synchrotron Radiation Centre SOLARIS' under contract number 1/SOL/2021/2 (project ID 233012, cryo-electron microscope). We thank M. Rawski, G. Ważny, M. Jaciuk and P. Indyka (all from SOLARIS Centre) for assistance in the preparation of grids and imaging. We thank J. Potempa (Jagiellonian University) for valuable discussions during the project and for financial support by providing salary to M.M. and covering the costs of materials and services from his financial resources (NCN grant UMO-2018/30/A/NZ5/00650) and others at the Jagiellonian University. B. Potempa (University of Louisville) is acknowledged for technical support. We thank T. Olczak (University of Wroclaw) for providing anti-HmuY antibodies. We thank O. Barczyk-Woznicka (Jagiellonian University) and M. Pacia (Jagiellonian University) for assistance with transmission electron microscopy experiments. The study was carried out using research infrastructure funded by the European Union in the framework of the Smart Growth Operational Programme, Measure 4.2; grant number POIR.04.02.00-00-D001/20, 'ATOMIN 2.0—Center for materials research on ATOMic scale for the INnovative economy'.

## Author contributions

A.S. determined cryo-EM structures, performed experiments in *B. thetaiotaomicron*, generated figures and wrote the paper. M.M. determined cryo-EM structures, generated figures and wrote the paper and performed experiments in *P. gingivalis* together with K.M. A.M.F. carried out proteomics, supervised by M.T. R.P.H. and A.J.H. carried out phylogenetic analyses, with A.J.H. supervised by R.P.H. A.B. managed the Newcastle University Structural Biology Facility. C.S. and J.J.E. carried out proteomics. B.v.d.B. conceived the project, performed experiments in *B. thetaiotaomicron* and wrote the paper.

## Competing interests

The authors declare no competing interests.

## Additional information

**Extended data** is available for this paper at https://doi.org/10.1038/s41564-025-02132-2.

**Correspondence and requests for materials** should be addressed to Mariusz Madej or Bert van den Berg.

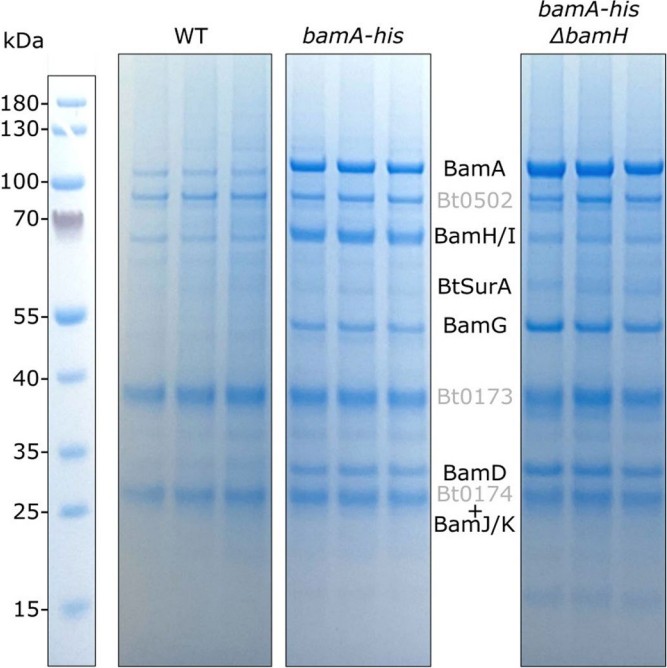

**Extended Data Fig. 1 | *B. thetaiotaomicron* pulldown sample SDS-PAGE analysis.** Coomassie-stained SDS-PAGE gel showing the protein profiles of IMAC pulldowns from *B. thetaiotaomicron* wild type, *bamA-his* and *bamA-his* + Δ*bamH* strains. The lanes correspond to independent biological repeats, which were analysed by quantitative proteomics. The proteomics results are shown in volcano plots in Fig. 1b and Fig. 4b. Prominent contaminants are annotated in grey.

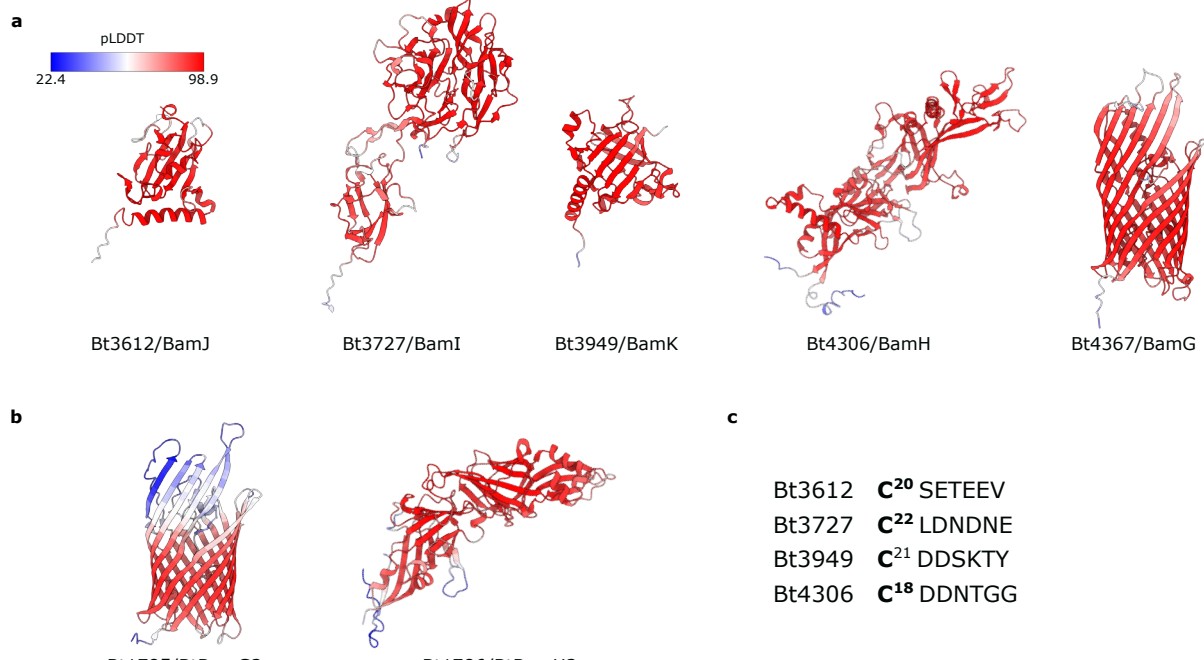

**Extended Data Fig. 2 | Structure predictions of proteins of interest. a**, AF2[29] predictions of abundant proteins identified in BtBamA$_{his}$ pulldowns. Protein models are coloured by pLDDT values (colour key). **b**, AF2 predictions of Bt1786 (BtBamH2) and Bt1785 (BtBamG2), which are homologues of Bt4306/BtBamH and Bt4367/BtBamG, respectively. Only the mature part of each protein is shown.

**c**, The lipid anchor cysteine and six subsequent amino acid residues of each lipoprotein detected in BtBamA$_{his}$ pulldowns. The presence of two or more acidic residues indicates that these lipoproteins have a lipoprotein export signal and are likely surface exposed[24,25].

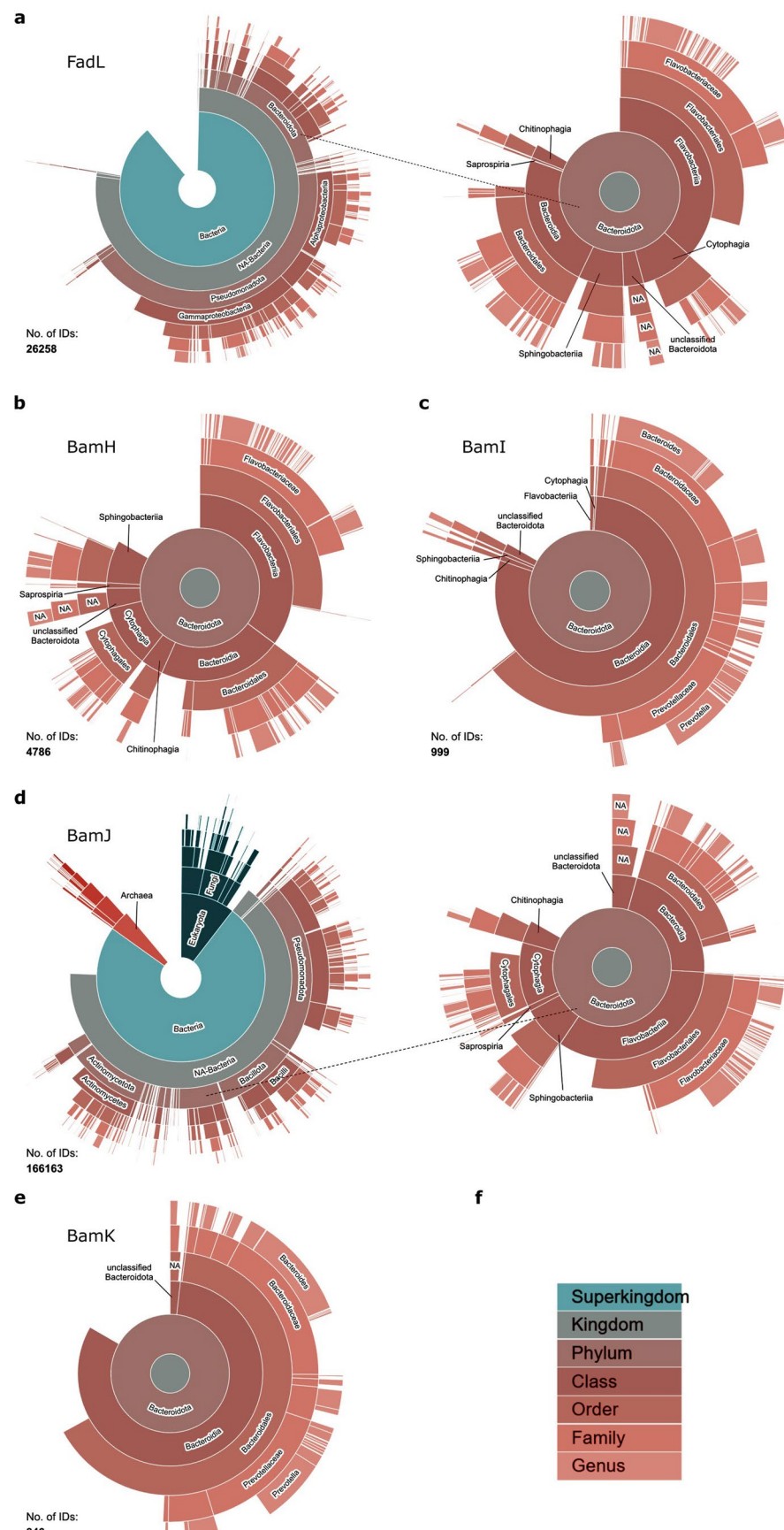

**Extended Data Fig. 3 | Distribution of BAM-associated surface-exposed lipoproteins in Bacteria.** Organisms containing homologues of BtBamG/FadL/ (**a**), BamH/Bt4306 (**b**), BamI/Bt3727 (**c**), BamJ/Bt3612 (**d**) and BamK/Bt3949 (**e**). **f**, Taxonomic level colour key. Displayed number of IDs corresponds to the number of pBLAST hits. The taxonomy sunburst plots were generated using the Enzyme Function Initiative - Enzyme Similarity Tool[79]. NA, unclassified.

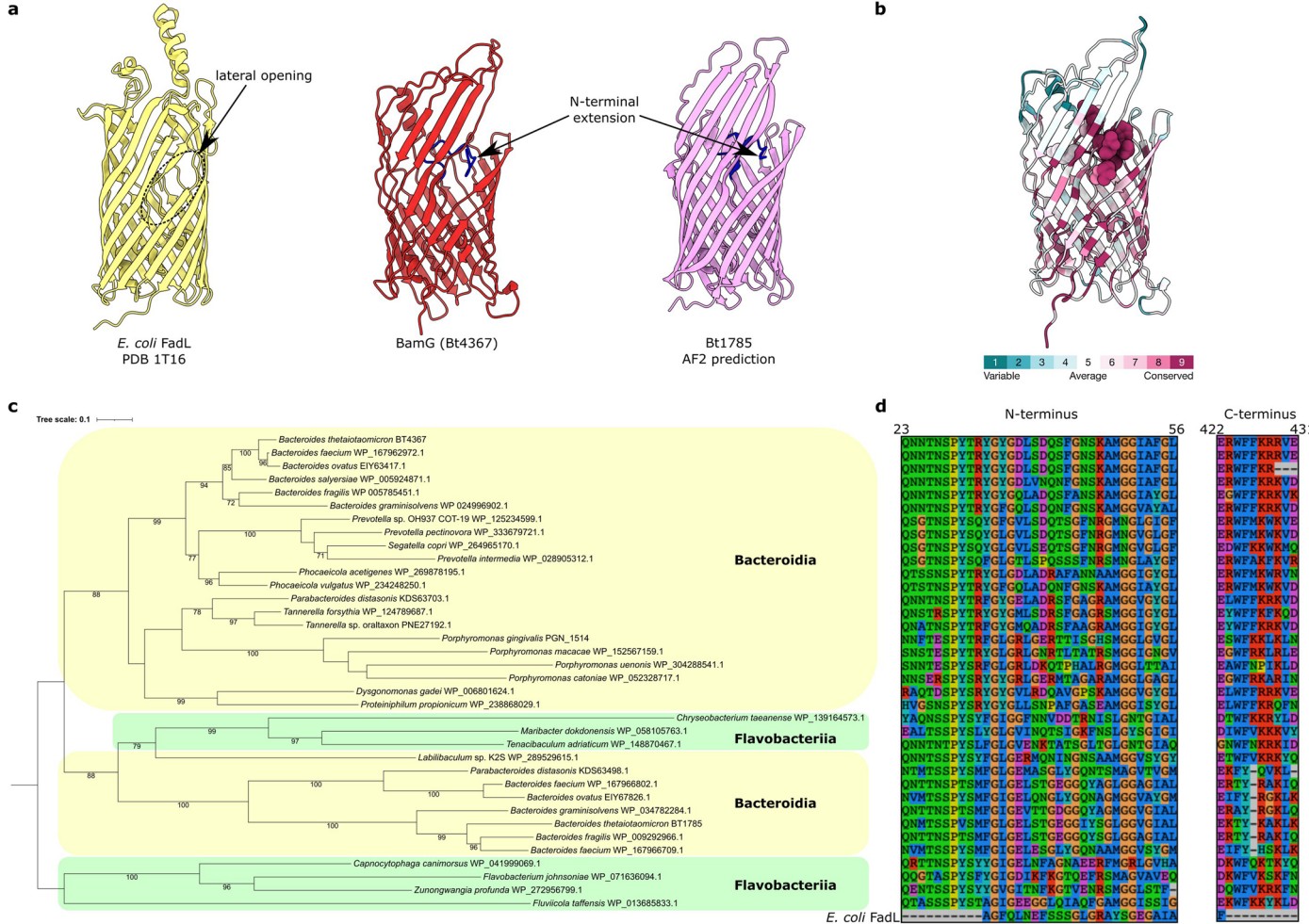

**Extended Data Fig. 4 | Properties and phylogenetics analysis of BamG.**
**a**, Comparison of the *E. coli* FadL crystal structure (PDB 1T16 ref. 31), the BtBamG cryo-EM structure and the AF2 prediction of BtBamG2/Bt1785 (signal sequence not shown). Views generated from a superposition. The 12-amino acid N-terminal extension of BtBamG and BtBamG2 is shown in blue. **b**, ConSurf analysis of BtBamG. The conserved N-terminal extension (residues 23–37) is shown in sphere representation. **c**, BamG protein phylogeny focusing on homologues encoded by selected members of the Bacteroidia and Flavobacteriia classes. The phylogeny was inferred from an alignment of 240 residues, and it is routed with a selection of members of the class Flavobacteriia, the closest outgroup to the Bacteroidia identified by phylogenomics[91]. The phylogeny was rooted with the lineage that includes the most basal branch among the Flavobacteriia, *Fluviicola taffensis*[91]. However, the phylogeny inferred from a broader taxa sampling does not resolve the position of the two distinct Flavobacteriia lineages (highlighted in green). The non-monophyly of the genus *Bacteroides* among the Bacteroidia (highlighted

in yellow) and Flavobacteriia could be due to a phylogenetic inference artefact or reflect a complex evolution of the BamG genes, including potential lateral gene transfers and/or gene duplications. Ultrafast bootstrap values (1000 replicates) below 95% indicate a lack of resolution for some of the branches to support either possibility. The strongly supported distinct positions for the two BamG homologues encoded by the species *Parabacteroides distasonis* (Family Tannerellaceae) suggest a lateral gene transfer from a *Bacteroides* species into *P. distasonis* (accession KDS63498.1). The scale bar indicates the inferred mutation per aligned residues for the automatically selected model (LG + G4 + I + F). The accession number for each protein in the alignment is indicated. **d**, Alignments of the N- and C-terminal regions of Bacteroidota BamG homologues showing conserved terminal extensions compared to *E. coli* FadL. Rows correspond to the sequences shown in the phylogeny in **c**; the bottom row shows the *E. coli* FadL sequence. Residue numbers at the top correspond to the BtBamG sequence.

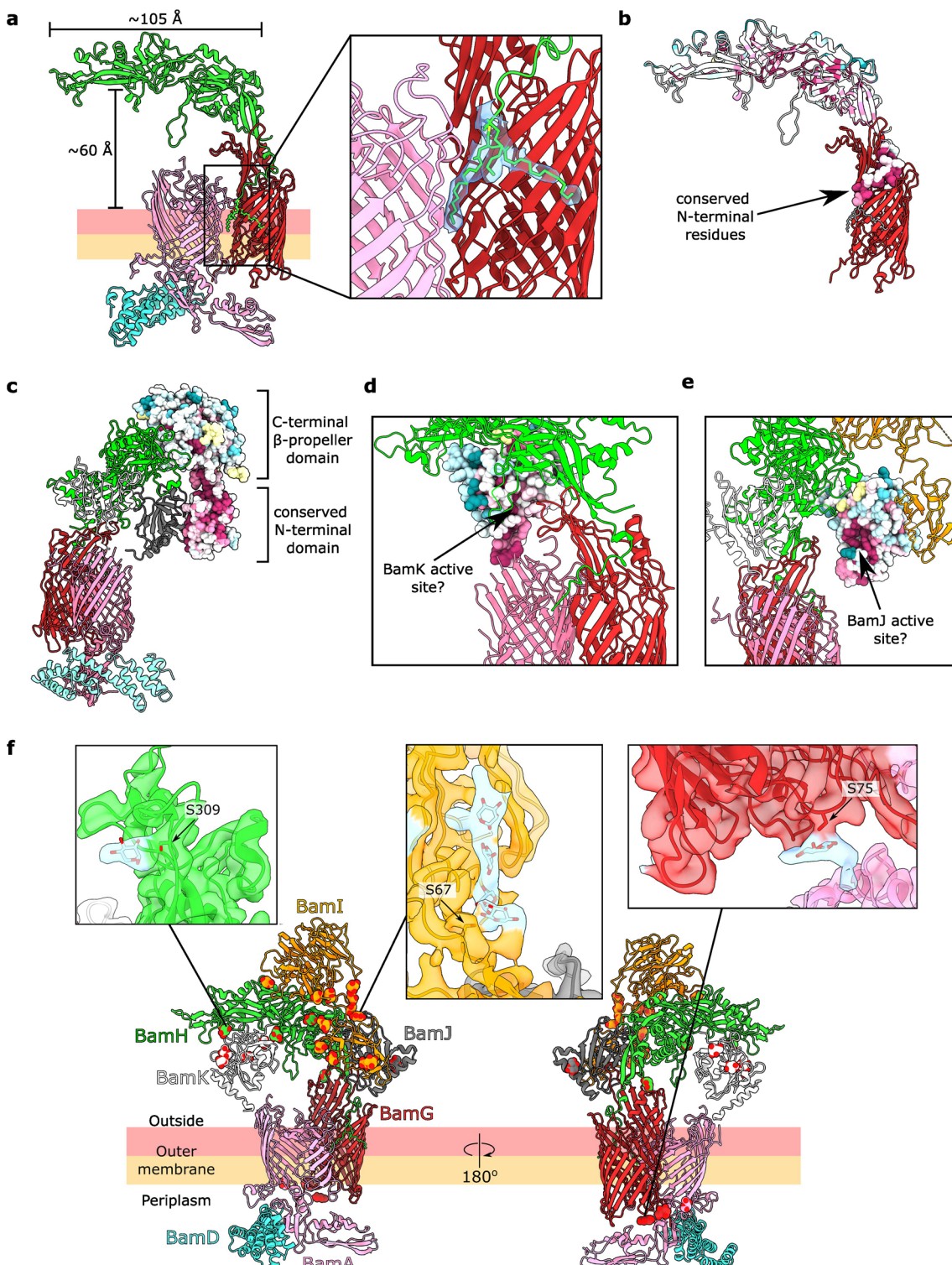

**Extended Data Fig. 5 | Properties and conservation analysis of previously undescribed BtBAM components. a**, BtBamH extends in an arch above the outer membrane, with its N-terminal cysteine lipid anchor wedged between BtBamA and BtBamG β-barrels. The inset shows the map-model fit of the lipid anchor. **b-e**, ConSurf[37] analyses of BtBamH (**b**), BtBamI (**c**), BtBamJ (**d**) and BtBamK (**e**), using default program parameters (minimum and maximum sequence identity set to 35% and 95%, respectively). Conserved BtBamH N-terminal residues 18–34 are shown in sphere representation in **b**. Putative BamK and BamJ PPI active sites face the interior of the extracellular dome in **d** and **e**. **f**, O-glycosylation sites displayed as mannose monosaccharides in space filling representation. The mannose residues were placed into the glycan densities to stabilise protein model refinement. The exact chemical nature of the *B. thetaiotaomicron* O-glycan is unknown. Insets show three representative glycans in blue extending from BamH Ser309, BamI Ser67, and BamG Ser75 sidechains.

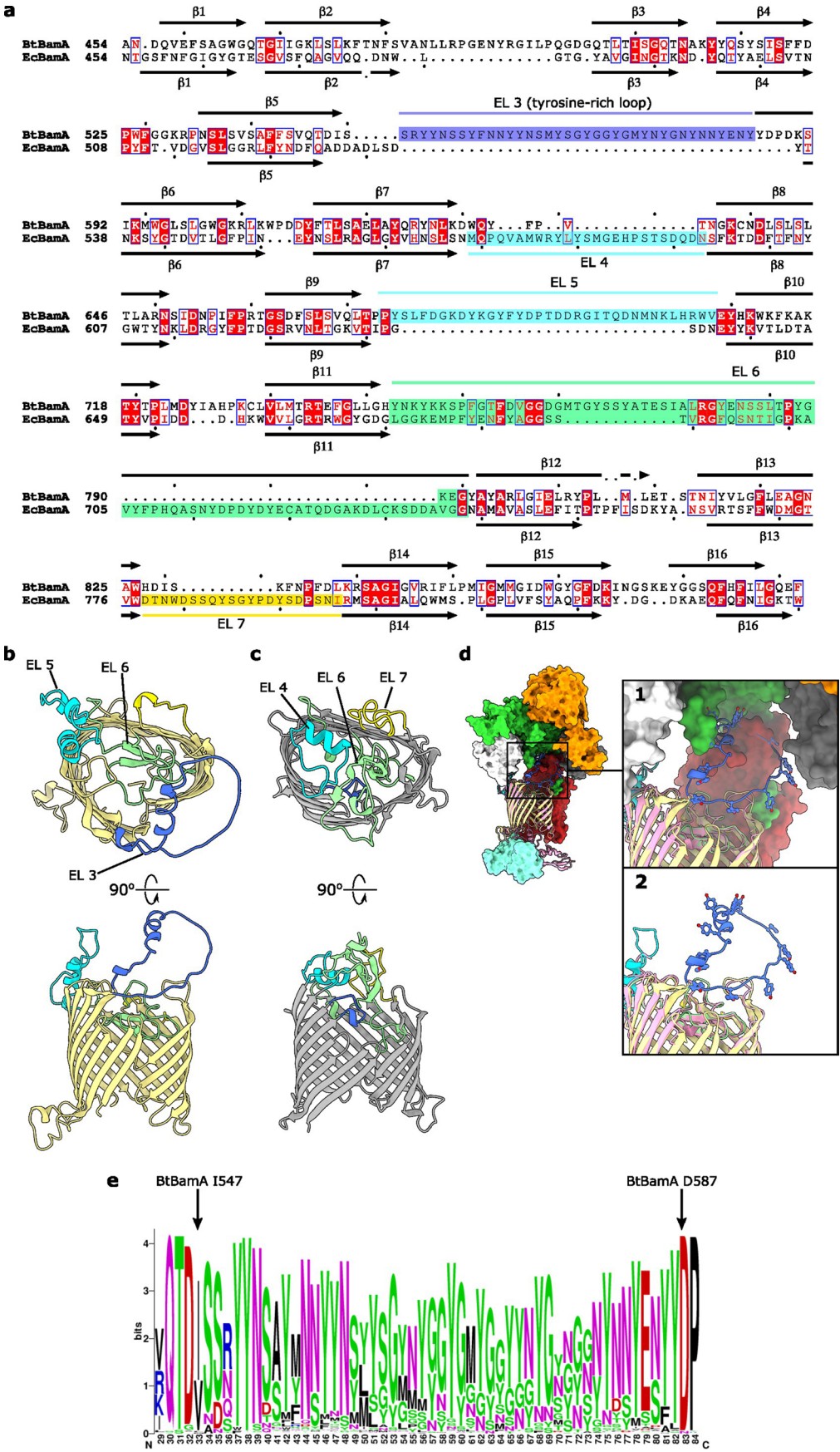

**Extended Data Fig. 6 | See next page for caption.**

**Extended Data Fig. 6 | Comparison of *B. thetaiotaomicron* and *E. coli* BamA β-barrels. a**, Structure-based sequence alignment of BtBamA and *E. coli* BamA (EcBamA). Sequence identity – 22%, TM-align score – 0.64. The β-barrel strands of BtBamA and EcBamA are annotated, respectively, above and below the sequence alignment. Extracellular loops (EL) of interest are annotated. **b, c**, AF2 models of BtBamA (**b**) and EcBamA (**c**) β-barrels with extracellular loops coloured as in (**a**). **d**, The position of the tyrosine-rich loop (EL 3) in the context of the whole BtBAM complex. The AF2 model of BtBamA, coloured as in **b**, is superposed on the experimental BtBamA structure. No cryo-EM density for the tyrosine-rich loop was observed. Inset 1 shows a close-up view of EL3 inside the dome. Inset 2 shows the same view of BtBamA as inset 1 with the other BAM components omitted for clarity. **e**, WebLogo[92] representation of the Bacteroidota BamA EL3 region generated from 861 aligned BlastP hits shows enrichment of tyrosine, glycine and asparagine residues.

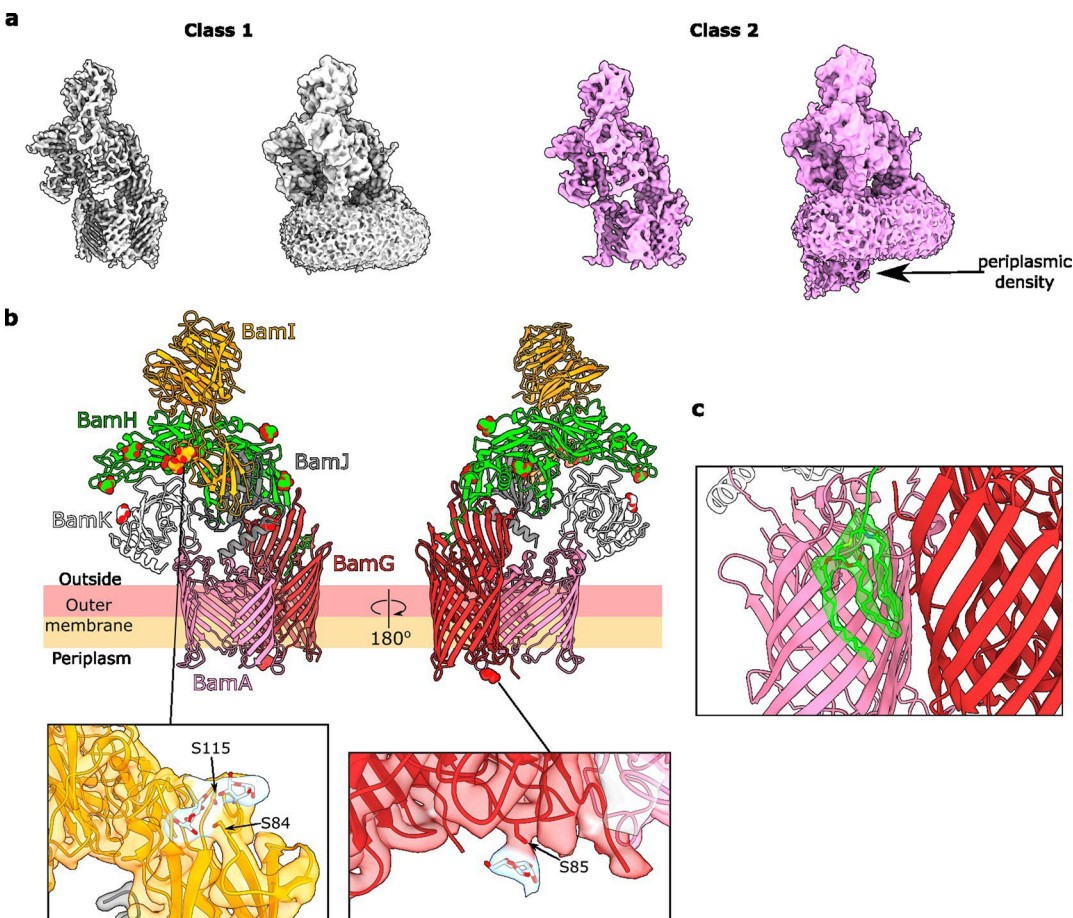

**Extended Data Fig. 7 | PgBAM structural features. a**, Two cryo-EM classes of PgBAM shown at high (left) and low (right) contour. The periplasmic density observed in class 2 is assumed to correspond to PgBamD and/or PgBamA POTRA domain 5. **b**, Glycosylation sites observed in the PgBAM structure. Mannose units (space filling representation) were built into the glycan density to stabilise protein model refinement. Insets show close-ups of representative model density at glycosylation sites on BamI (Ser84, Ser115) and BamG (Ser85). **c**, Model-density fit of the PgBamH lipid anchor at the PgBamAG barrel interface.

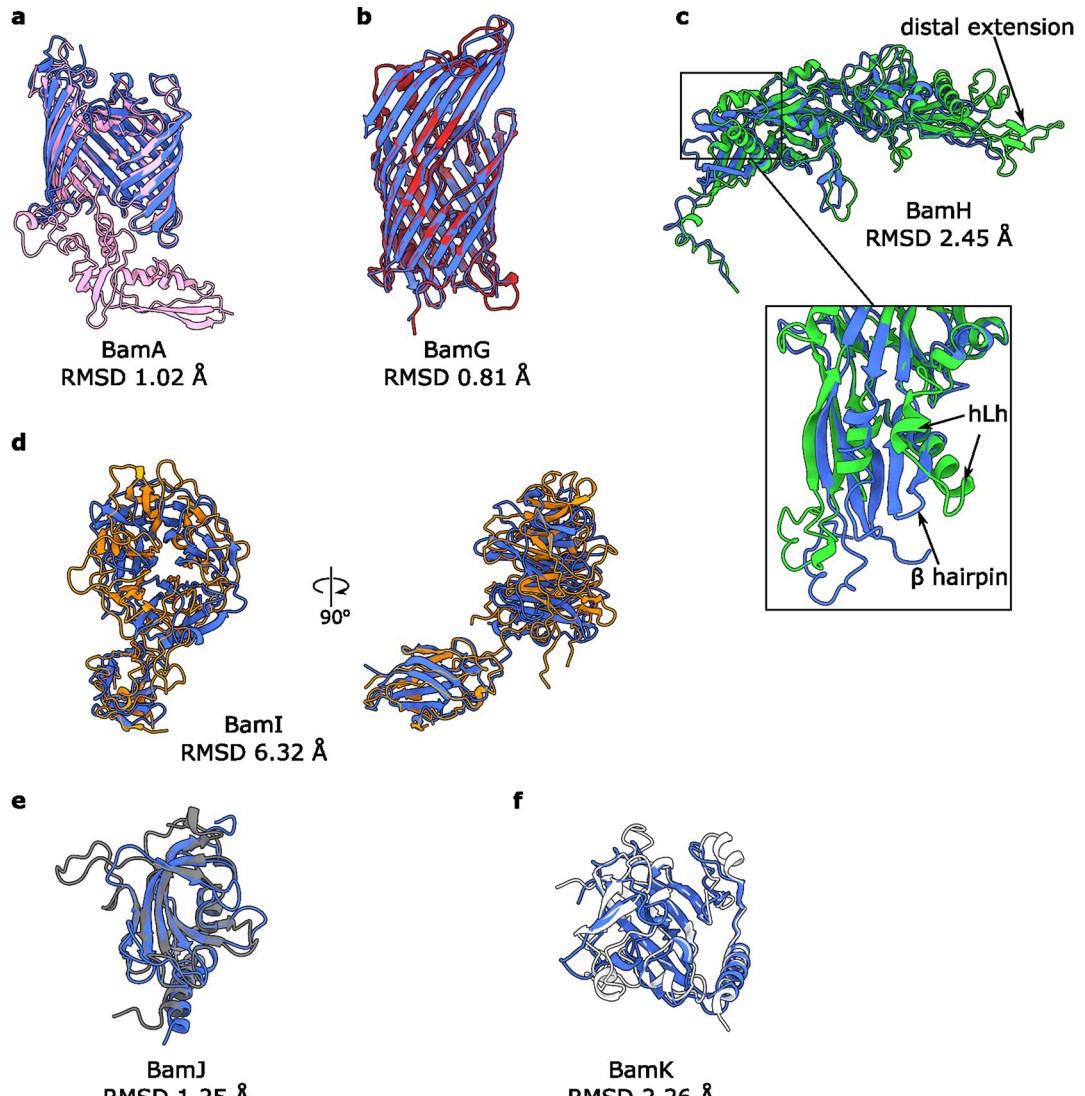

**Extended Data Fig. 8 | Structural alignment of individual BtBAM and PgBAM components.** Superpositions of BamA (**a**), BamG (**b**), BamH (**c**), BamI (**d**), BamJ (**e**) and BamK (**f**) models from *B. thetaiotaomicron* and *P. gingivalis* BAM complexes. Models were aligned using Matchmaker in ChimeraX[93], and the Cα-Cα RMSD between pruned atom pairs is reported. BtBAM components are coloured using the same colour scheme as in Fig. 2 of the main text; PgBAM components are shown in blue. hLh, helix-loop-helix.

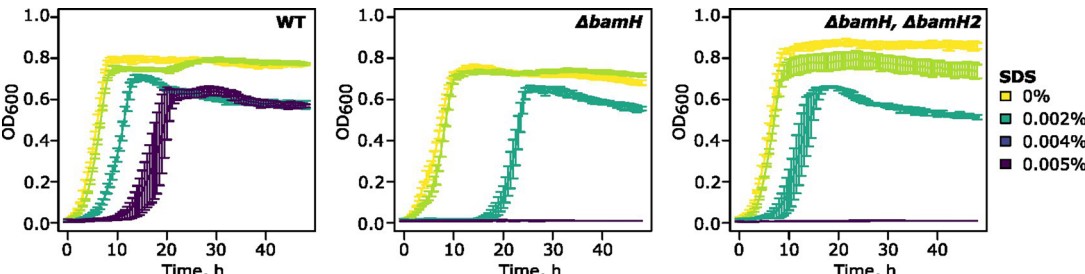

**Extended Data Fig. 9 | Growth of *B. theta bamH* knockouts in the presence of SDS.** Growth of wild type, Δ*bamH* and Δ*bamH*, Δ*bamH2 B. thetaiotaomicron* strains in EBHI in the presence of indicated concentrations of sodium dodecyl sulphate (SDS. Each trace is an average of n = 3 technical repeats; the error bars represent the SD. Curves shown are representative of two biological repeats done on different days.

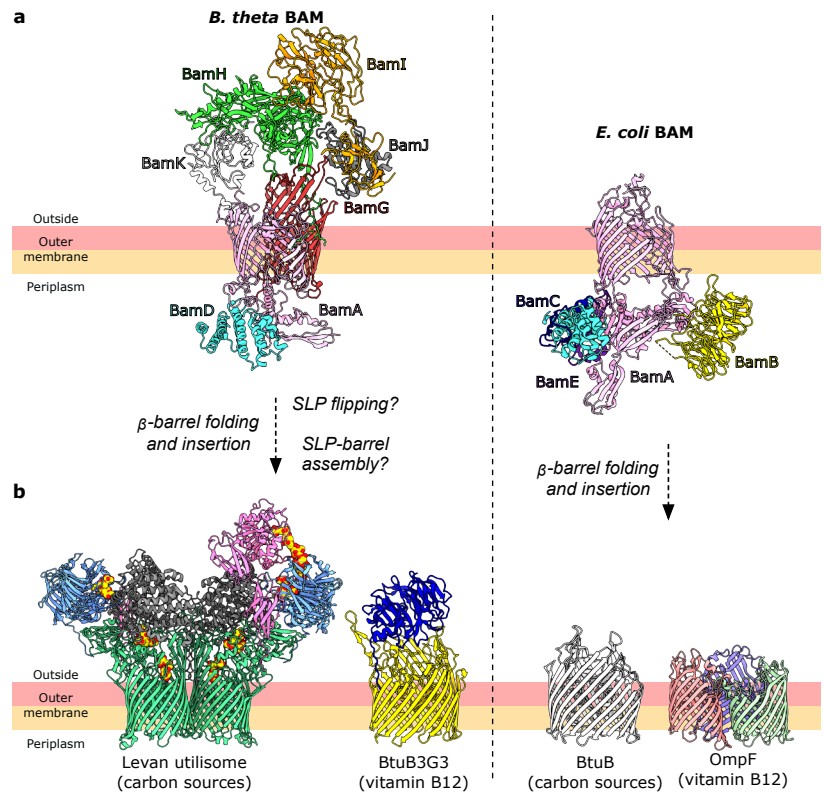

**Extended Data Fig. 10 | Comparison of *B. theta* and *E. coli* BAM complexes and their typical substrates. a**, Comparison of BtBAM (PDB 9HJ3) and EcBAM (PDB 5DOO[3]) structures. The views were generated from a superposition of the BamA chains. **b**, Examples of *B. theta* and *E. coli* BAM substrates: levan utilisome (Bt1760-3), PDB 8AA2 ref. 46; vitamin $B_{12}$ transporter BtuB3G3 (Bt2094-5), PDB 8P98 ref. 94; vitamin B12 transporter BtuB, PDB 1NQE ref. 95; general porin OmpF, PDB 1OPF ref. 96. As is clear from the comparison of OM transporters for carbon sources and vitamin $B_{12}$, the Bacteroidota BAM complex likely requires expanded functionality compared to its Proteobacterial counterpart.

# Reporting Summary

## Statistics

For all statistical analyses, confirm that the following items are present in the figure legend, table legend, main text, or Methods section.

| n/a | Confirmed | |
|---|---|---|
| ☐ | ☒ | The exact sample size (*n*) for each experimental group/condition, given as a discrete number and unit of measurement |
| ☐ | ☒ | A statement on whether measurements were taken from distinct samples or whether the same sample was measured repeatedly |
| ☐ | ☒ | The statistical test(s) used AND whether they are one- or two-sided<br>*Only common tests should be described solely by name; describe more complex techniques in the Methods section.* |
| ☒ | ☐ | A description of all covariates tested |
| ☒ | ☐ | A description of any assumptions or corrections, such as tests of normality and adjustment for multiple comparisons |
| ☐ | ☒ | A full description of the statistical parameters including central tendency (e.g. means) or other basic estimates (e.g. regression coefficient) AND variation (e.g. standard deviation) or associated estimates of uncertainty (e.g. confidence intervals) |
| ☐ | ☒ | For null hypothesis testing, the test statistic (e.g. *F*, *t*, *r*) with confidence intervals, effect sizes, degrees of freedom and *P* value noted<br>*Give P values as exact values whenever suitable.* |
| ☒ | ☐ | For Bayesian analysis, information on the choice of priors and Markov chain Monte Carlo settings |
| ☒ | ☐ | For hierarchical and complex designs, identification of the appropriate level for tests and full reporting of outcomes |
| ☒ | ☐ | Estimates of effect sizes (e.g. Cohen's *d*, Pearson's *r*), indicating how they were calculated |

*Our web collection on statistics for biologists contains articles on many of the points above.*

## Software and code

Policy information about availability of computer code

| | |
|---|---|
| Data collection | Gatan Microscopy Suite v3.4.3, Evosep One HyStar Driver 2.3.57.0, Compass HyStar v6.2, BLASTP 2.12.0+ |
| Data analysis | cryoSPARC v4.5.3, Coot v0.9.8.8, Phenix v1.20.1-4487, AlphaFold v2, AlphaFold v3 (web server), UCSF ChimeraX v1.6-v1.9, ConSurf (web server), RStudio 2024.12.0+467, Microsoft Excel 365, DIA-NN v1.8.2.27, KNIME v5.1.2, FragPipe-Analyst, Limma, Enzyme Function Initiative - Enzyme Similarity Tool, SeaView v5.0.5, IQ-TREE. |

For manuscripts utilizing custom algorithms or software that are central to the research but not yet described in published literature, software must be made available to editors and reviewers. We strongly encourage code deposition in a community repository (e.g. GitHub). See the Nature Portfolio guidelines for submitting code & software for further information.

## Data

Policy information about availability of data

All manuscripts must include a data availability statement. This statement should provide the following information, where applicable:

- Accession codes, unique identifiers, or web links for publicly available datasets
- A description of any restrictions on data availability
- For clinical datasets or third party data, please ensure that the statement adheres to our policy

Pulldown proteomic mass spectrometry data sets and results files are referenced in ProteomeXchange (PXD058163) and available to download from MassIVE (MSV000096498) [doi:10.25345/C57W67J00]. Pre-publication access can be obtained with the following link ftp://MSV000096498@massive.ucsd.edu (username

MSV000096498, password S73xCTORvm9SM2m7). B. theta membrane and P. gingivalis whole cell mass spectrometry proteomics data have been deposited to the ProteomeXchange Consortium via the PRIDE partner repository with the dataset identifiers PXD058903 for B. theta and PXD058905 for P. gingivalis. (reviewers can access the dataset by logging in to the PRIDE website using the following account details for B. theta: username: reviewer_pxd058903@ebi.ac.uk,password afLrMXZevbBq; or P. gingivalis: project accession: PXD058905, token: eAL4YgXN00IJ. Cryo-EM reconstructions and atomic coordinate files with the indicated accession numbers have been uploaded to the Electron Microscopy Data Bank (EMDB) and the Protein Data Bank (PDB): BtBAM class 1 (EMD-52200 and 9HIS), BtBAM class 2 (EMD-52202 and 9HIV), BtBAM composite map and model (EMD-52209 and 9HJ3), PgBAM class 1 (EMD-52218 and 9HJM), and PgBAM class 2 (EMD-52219).

# Research involving human participants, their data, or biological material

Policy information about studies with [human participants or human data](). See also policy information about [sex, gender (identity/presentation), and sexual orientation]() and [race, ethnicity and racism]().

| | |
|---|---|
| Reporting on sex and gender | N/A |
| Reporting on race, ethnicity, or other socially relevant groupings | N/A |
| Population characteristics | N/A |
| Recruitment | N/A |
| Ethics oversight | N/A |

Note that full information on the approval of the study protocol must also be provided in the manuscript.

# Field-specific reporting

Please select the one below that is the best fit for your research. If you are not sure, read the appropriate sections before making your selection.

☒ Life sciences      ☐ Behavioural & social sciences      ☐ Ecological, evolutionary & environmental sciences

For a reference copy of the document with all sections, see [nature.com/documents/nr-reporting-summary-flat.pdf](http://nature.com/documents/nr-reporting-summary-flat.pdf)

# Life sciences study design

All studies must disclose on these points even when the disclosure is negative.

| | |
|---|---|
| Sample size | No statistical methods were used to predetermine sample sizes. Sample sizes were set to 2 or 3 (or more) to assess and ensure standard reproducibility.<br>For cryo-EM, the collected datasets contained enough particles for structure determination. |
| Data exclusions | Cryo-EM movies with unacceptable CTF parameters, average intensity and relative ice thickness were excluded from data analysis. Particle images that did not yield good 2D class averages or 3D reconstructions were excluded from single particle analysis. |
| Replication | Structure determination experiments were not repeated, as is standard practice in the field.<br>Biochemical and growth assays were repeated at least three times on different days to ensure reproducibility. |
| Randomization | Particle stacks were randomly split into two independent sets during 3D volume reconstruction in single particle analysis.<br>For all other experiments, randomization was not possible. |
| Blinding | Knowing the identities of the samples was required for carrying out the analyses, and no blinding was performed as part of data analysis.<br>Investigators were not blinded to group allocation, as no grouping was needed for this study. |

# Reporting for specific materials, systems and methods

We require information from authors about some types of materials, experimental systems and methods used in many studies. Here, indicate whether each material, system or method listed is relevant to your study. If you are not sure if a list item applies to your research, read the appropriate section before selecting a response.

## Materials & experimental systems

| n/a | Involved in the study |
|:---:|:---|
| ☐ | ☒ Antibodies |
| ☒ | ☐ Eukaryotic cell lines |
| ☒ | ☐ Palaeontology and archaeology |
| ☒ | ☐ Animals and other organisms |
| ☒ | ☐ Clinical data |
| ☒ | ☐ Dual use research of concern |
| ☒ | ☐ Plants |

## Methods

| n/a | Involved in the study |
|:---:|:---|
| ☒ | ☐ ChIP-seq |
| ☒ | ☐ Flow cytometry |
| ☒ | ☐ MRI-based neuroimaging |

## Antibodies

| | |
|:---|:---|
| Antibodies used | anti-HmuY polyclonal antibodies (concentration used- 1 μg/ml)  - provided by Prof. Teresa Olczak (University of Wroclaw, Poland), HRP-conjugated anti-rabbit antibodies (Sigma Aldrich, A0545) - dilution 1:20,000 |
| Validation | anti-HmuY polyclonal antibodies were valideted in published manuscript: doi: 10.1371/journal.pone.0117508. eCollection 2015 HRP-conjugated anti-rabbit antibodies (Sigma Aldrich, A0545) were used according to manufacturer's recommendations. |

## Plants

| | |
|:---|:---|
| Seed stocks | *Report on the source of all seed stocks or other plant material used. If applicable, state the seed stock centre and catalogue number. If plant specimens were collected from the field, describe the collection location, date and sampling procedures.* |
| Novel plant genotypes | *Describe the methods by which all novel plant genotypes were produced. This includes those generated by transgenic approaches, gene editing, chemical/radiation-based mutagenesis and hybridization. For transgenic lines, describe the transformation method, the number of independent lines analyzed and the generation upon which experiments were performed. For gene-edited lines, describe the editor used, the endogenous sequence targeted for editing, the targeting guide RNA sequence (if applicable) and how the editor was applied.* |
| Authentication | *Describe any authentication procedures for each seed stock used or novel genotype generated. Describe any experiments used to assess the effect of a mutation and, where applicable, how potential secondary effects (e.g. second site T-DNA insertions, mosiacism, off-target gene editing) were examined.* |

