## [Peer Review File · Nature Microbiology]

Structure of a distinct BAM complex in the Bacteroidota

Corresponding Author: Professor Bert van den Berg

Version 0:

Reviewer comments:

Reviewer #1

(Remarks to the Author)

The authors have done a great job addressing the previous comments. I find all my issues resolved satisfactorily. The work and the results are exciting and groundbreaking and highly deserve publication in Nature Microbiology.

Minor issue: Please check carefully the shifted nomenclature, I spotted in lines 320-330 several instances where BamG should be changed to BamH. There might be other instances elsewhere.

Decision Letter:

Our ref: NMICROBIOL-25062258-T

28th July 2025

Dear Dr. van den Berg,

Thank you for submitting your revised manuscript "Discovery of a distinct BAM complex in the Bacteroidetes" (NMICROBIOL-25062258-T). It has now been seen by the original referees and their comments are below. The reviewers find that the paper has improved in revision, and therefore we'll be happy in principle to publish it in Nature Microbiology, pending minor revisions to satisfy the referees' final requests and to comply with our editorial and formatting guidelines.

We are now performing detailed checks on your paper and will send you a checklist detailing our editorial and formatting requirements in one to two weeks. Please do not upload the final materials and make any revisions until you receive this additional information from us.

Thank you again for your interest in Nature Microbiology Please do not hesitate to contact me if you have any questions.

Sincerely,

Reviewer #1 (Remarks to the Author):

The authors have done a great job addressing the previous comments. I find all my issues resolved satisfactorily. The work and the results are exciting and groundbreaking and highly deserve publication in Nature Microbiology.

Minor issue: Please check carefully the shifted nomenclature, I spotted in lines 320-330 several instances where BamG should be changed to BamH. There might be other instances elsewhere.

Version 1:

Decision Letter:

21st August 2025

Dear Professor van den Berg,

I am pleased to accept your Article "Structure of a distinct BAM complex in the Bacteroidota" for publication in Nature Microbiology. Thank you for having chosen to submit your work to us and many congratulations.

You may wish to make your media relations office aware of your accepted publication, in case they consider it appropriate to organize some internal or external publicity. Once your paper has been scheduled you will receive an email confirming the publication details. This is normally 3-4 working days in advance of publication. If you need additional notice of the date and time of publication, please let the production team know when you receive the proof of your article to ensure there is sufficient time to coordinate. Further information on our embargo policies can be found here:

<https://www.nature.com/authors/policies/embargo.html>

Authors may need to take specific actions to achieve compliance with funder and institutional open access mandates. If

your research is supported by a funder that requires immediate open access (e.g. according to [Plan S principles](https://www.springernature.com/gp/open-science/plan-s-compliance) or the [NIH public access policy](https://www.springernature.com/gp/open-science/us-federal-agency-compliance)) then you should select the gold OA route, and we will direct you to the compliant route where possible. Because authors warrant under our subscription licensing terms that they haven't committed to licensing any version of their article under a licence inconsistent with the terms of our agreement – including the applicable embargo period – publication under the subscription model isn't suitable for authors whose funders require no embargo.

We welcome the submission of potential cover material (including a short caption of around 40 words) related to your manuscript; suggestions should be sent to Nature Microbiology as electronic files (the image should be 300 dpi at 210 x 297 mm in either TIFF or JPEG format). Please note that such pictures should be selected more for their aesthetic appeal than for their scientific content, and that colour images work better than black and white or grayscale images. Please do not try to design a cover with the Nature Microbiology logo etc., and please do not submit composites of images related to your work, or AI-generated images. I am sure you will understand that we cannot make any promise as to whether any of your suggestions might be selected for the cover of the journal.

With kind regards,

P.S. Click on the following link if you would like to recommend Nature Microbiology to your librarian
<http://www.nature.com/subscriptions/recommend.html#forms>

** Visit the Springer Nature Editorial and Publishing website at http://editorial-jobs.springernature.com?utm_source=ejP_NMicro_email&utm_medium=ejP_NMicro_email&utm_campaign=ejp_NMicro for more information about our career opportunities. If you have any questions please click [here](mailto:editorial.publishing.jobs@springernature.com).

Referee #1 (Remarks to the Author):

The beta-barrel assembly machinery (Bam complex) is central to bacterial outer membrane biogenesis. In this manuscript, Silale and co-authors report that the Bam complex in two Bacteroidales species, *Bacteroides thetaiotaomicron* and *Porphyromonas gingivalis*, contains a conserved BamA-BamD core, along with unique components. These include an additional essential beta-barrel protein (BamF) and four associated surface-exposed lipoproteins (SLPs) named BamG–J. This represents a significant deviation from the Bam complex paradigm in Proteobacteria, where accessory lipoproteins are typically located in the periplasm.

The significance of this study lies in its emphasis on the value of investigating outer membrane biogenesis across diverse diderm bacteria. By identifying both conserved and variable components of the Bam complex, this work encourages comparative studies of envelope assembly across species. **However, the proposed role of these novel components—particularly in the assembly of SLP-barrel complexes—remains speculative and is not supported by direct functional evidence.**

Major Comments:

1. The name BamF has already been used to describe a unique component of the Bam complex in α -Proteobacteria. I strongly encourage the authors to reconsider this naming choice to avoid confusion in the field, particularly regarding the conservation of Bam complex components.

We have renamed all the components in the most straightforward fashion by moving all the names up (e.g. BamF becomes BamG, BamG becomes BamH, etc). Please note we will use the new naming in the rest of our rebuttal.

2. The study reports that bamF is likely essential, and that bamG mutants in both species exhibit pleiotropic growth and envelope-related phenotypes. However, there is no direct evidence linking these components to the core Bam complex. It is possible that BamF and the associated lipoproteins have independent functions, especially considering that BamF is a close homologue of FadL.

We assume the reviewer refers to the lack of evidence functionally linking the new components to BamAD, rather than physically (which we think is clear based on the structures). We consider it very unlikely that BamG–K have independent functions given their stable association (via BamG) with BamA in three different organisms (including Liu et al.). For example, the dramatic growth defects of *P. gingivalis* delta BamH/I/J on MM + BSA (Fig. 4e) can be explained by assembly defects of the RagAB complex (Fig. 5f–h), which is essential for import of peptides in *P. gingivalis*.

With respect to BamG, this is indeed a FadL homologue. However, as we explain in the text, the fact that the lateral opening of the BamG barrel is occupied by the N-termini of both BamG itself and BamH seems to preclude a FadL-like function.

3. The BtBAM complex appears to co-purify with numerous other proteins, as shown in Figures 1 and 4 (in the absence of BamG). However, the mass spectrometry data underlying the volcano plot are not included in the Supplemental Materials, and I believe they should be provided for the reader's convenience.

We agree that this is useful and have included the mass spectrometry data from Figures 1b and 4b as requested. This is the new Supplementary Table 1. Since we wanted to ensure capturing all potential BAM-interacting proteins, we didn't wash stringently during IMAC. As a result, there are many background hits, which is now clear from the added Supplementary Table 1.

While the authors label and follow up on a select few hits, it is unclear why these particular proteins were chosen over others as putative components of the Bam complex. I presume the selection was based on the success of solving the structure of the complex, but it is also possible that these proteins are simply substrates, chaperones, or other factors that co-purified with BamA/D due to relative abundance, rather than being bona fide functional components of the Bam complex.

The choice of proteins was based on abundance in the proteomics and their copurification with BamA following SEC as shown in Fig. 1a. In addition, the BAM complexes are very stable in cryoEM, as no subcomplexes were observed. Thus, we consider it very unlikely that the novel components (BamG-K) are not an integral part of the BAM complex. This is supported by SurA (Bt3848). In Pseudomonadota, the chaperone SurA interacts with the BAM complex and has recently been co-purified with EcBAM under carefully controlled conditions (e.g. PMID: 40184469). In our proteomics, BtSurA is very abundant following pulldown (Figs. 1b and 4b and Supplementary Table 1), but it is lost following SEC (Fig. 1a), distinguishing it from authentic BAM subunits. As is clear from the new Supplementary Table 1, there are other potential BAM-interacting proteins, such as Bt4261 and Bt3222, both of which are SLPs. Exploring the function of these proteins is beyond the scope of this work.

4. Similarly, the proposed role of these components in the biogenesis of SLP-barrel complexes is not well-supported by the current data. Many of the observed changes could be due to differential gene expression or indirect downstream effects considering highly pleiotropic phenotypes. Substantial additional work would be needed to establish the specific functions of these novel components, so I suggest to tone down this claim. We agree with the reviewer to a certain extent.

As mentioned in the text, the fact that BamH is extracellular limits the potential of the deletion mutant to directly interfere with intracellular processes. Given this, we think our data shows that the *DbamH* mutant causes general OM permeability defects and leads

to the disappearance of some OMP (SLP-barrel) complexes, including crucial ones for *P. gingivalis*. Importantly, unlike the depletion experiments by Liu et al., the *bamH* deletion does not affect the levels of BamA and BamG, and we have therefore uncoupled the effect of removal of the extracellular dome from general defects that would be caused by lower levels of BamA and BamG. Our *bamH* deletion strain appears similar to the *bamH* suppressor strain under control of a weak promoter reported by Liu et al.

We agree that, as for Liu et al., specific roles cannot yet be assigned to the newly discovered components, and hope that the reviewer agrees that this lies beyond the scope of the current study. We have therefore toned down our speculative claims regarding function.

Additional comments:

1. The idea that the Bam complex may contain unique components with specialized functions is not unprecedented. In *E. coli*, Bam lipoproteins affect different substrates, and other Proteobacteria exhibit distinct Bam complex compositions, including unique components such as BamF in α -Proteobacteria. Similarly, mitochondrial homologs of BamA also have specific partner proteins on the mitochondrial surface. An expanded discussion of these examples could help place the current findings into a broader biological context.

We thank the reviewer for this suggestion. However, the comment about "other Proteobacteria exhibit distinct Bam complex compositions" is a bit misleading. As far as we're aware, BamF is the only known distinct bacterial Bam protein. It has a very narrow distribution and might have a BamC-like function.

We have replaced the functional speculation in the discussion with a brief overview of additional BAM components (lines 405-415).

2. The supplemental data table with the proteomic analysis of protein abundance would be significantly more valuable if it included functional annotations of the identified proteins, as well as indications of which are predicted lipoproteins or beta-barrel OMPs.

We agree and have included this information (Supplementary Tables 4 and 5).

3. "Deletion of *bamG* disrupts the main SLP-barrel complexes of *P. gingivalis*". The data presented in this section are inconclusive, since the RagA protein is not detected in the analysis.

RagA is in fact present in the *bamH* deletion strain and is also present (at low levels) in the sample after purification of RagBhis from this strain. We don't think these data are inconclusive. In previous work (PMC7610489; Fig. 4b) and stated in our manuscript (lines 381-2), we have shown that deletion of the RagB SLP (disrupting the RagAB complex) leads to RagA loss in the OM. Given that RagB is present, disruption of the RagAB

complex resulting from BamH removal is the most likely explanation for the strong phenotypes.

4. Lines 449–454: This statement is highly speculative. The absence of an essential OMP does not imply that BamA functions as an SLP flippase. It is possible that individual SLPs depend on specific, non-essential OMPs for their surface localization—similar to how SLAM proteins or the RcsF/OMP complexes function. Therefore, the non-essentiality of individual OMPs does not rule out their specialized roles in SLP translocation. **We agree and have removed this section to avoid speculation on specific function(s) of the new BAM components. This includes the old ED Fig. 9.**

5. Lines 489–490: I caution against overinterpreting the AlphaFold prediction of a Bam/SusCD supercomplex (ED Fig. 10). This predicted interaction may simply result from hydrophobic contacts between barrel surfaces in the absence of a membrane environment, rather than reflecting a biologically relevant complex.

This could indeed be the case (and admittedly the predictions are low-confidence), but AlphaFold predictions are not driven by hydrophobic matching but by evolutionary conservation. Also, if the interaction was driven by hydrophobicity, SusCD would be expected to bind to the BamAG interface, where the interacting surface would be larger. Thus, the fact that SusCD is always modelled in the same place at the front of BamA might be relevant. However, we have toned down our speculation on functional implications and have removed the old ED Fig. 10.

Referee #2 (Remarks to the Author):

The manuscript by Silale, Madej et al describes the identification, isolation and structure determination of two BAM complexes from *Bacteroides* and the partial functional characterization of one subunit by genetics and proteomics. The manuscript is generally very well written and the data clearly accessible (with few exceptions, see below). The work is technically sound, an inspection of the EM maps and models showed no obvious irregularities. The work is groundbreaking, as a fundamentally new membrane protein complex has been found and its structure determined. The impact is maximal from multiple angles, including antibiotic resistance, membrane protein biogenesis and evolution. The findings are thus highly exciting not only for a Structural Biology and Microbiology audience, but also for a general audience. We congratulate the authors to this achievement.

We thank the reviewers for their kind comments, and we appreciate their recognition of the significance of our study.

We highly support publication of the manuscript in *Nature*, after the following issues have been satisfactorily addressed:

Nomenclature comments:

- The manuscript has been submitted back-to-back with another manuscript describing other Bacteroides complexes. It seems that the nomenclature of the two manuscripts is in agreement with each other, but it is unclear to the reader whether this is coincidence or purpose (BamG seems to correspond to BamG, the other subunits are different and also have different names). I strongly recommend that in both manuscripts, a short statement is included that the nomenclatures have been matched, to avoid any confusion with regard to this important issue. This will be particularly important for readers new to the field.

Given that the papers won't be published back-to-back in Nature, we have not (yet) addressed this issue but may address it during the proofreading stage. The nomenclature has been matched.

- Along the same lines, the name "BamF" has however already been given to a protein associated with the BAM complex (Anwari et al. Mol. Microbiol. 84, 832 (2012)), a BAM subunit of α -proteobacterium *Caulobacter crescentus*. Such confusion of nomenclature should be avoided. Therefore, the authors should either revise their nomenclature, or, mention the other manuscript and explain why they would override that name. This needs also be coordinated with the back-to-back manuscript.

We were aware of the existence of BamF from *Caulobacter*, and the manuscript was cited in our paper (original ref. 53, now ref. 28). We reasoned that the phylogenetic separation would allow for another protein named BamF. However, we now feel it is better to avoid any confusion, and have therefore, in coordination with the other manuscript, decided to move all names of the new subunits up one letter, e.g. BamF becomes BamG, BamG becomes BamH, etc. As stated above, in our response we have used the new names.

- The manuscript uses up to three different names for the various proteins and genes, Bt_XXXX, BamXX, and sometimes DUFXXX. This makes it hard to read. I strongly recommend to switch the entire manuscript consistently to the BamXX names and give a supplementary or ED Table with the other names. We have now used BamXX names in the manuscript.

Major comments:

- Fig. 4c, d and Supp. Fig. 4 need to include error bars and/or shading the bandwidth of the replicates. It is currently unclear whether the difference between wt and knock-out is significant.

We have made these changes. The differences are clearly significant.

- A knockout of BamG appears to be viable. But this does not necessarily mean that BamG is not essential. A reasonable alternative could be that BamG is functionally essential but is replaced by one of its paralogues (isoforms). This possibility is ruled out by the authors, because the alternative BamG forms were not detected by proteomics. There can be many reasons why any given protein is not detected in mass spec, so this

explanation falls short of a strict statement. The experiment on BamG essentiality should therefore be repeated by knocking out all BamG paralogues. Given the importance of BamG in the architecture of the BAM complex, and given that the back-to-back manuscript finds that BamG is essential in their species, so this is an important aspect. There's only one paralog of BamH in *B. theta* (BamH2; Bt1786). While the reviewers are correct that the absence of a protein in proteomics can have several causes, we note that BtBamH and the other BAM components are relatively abundant. Coupled to the sensitivity of modern mass spectrometers, this makes it very unlikely that we couldn't detect BamH2 (and BamG2; Bt1785) at a level that would make complementation with BAM meaningful. However, we have removed *bamH2* within the *DbamH* background, and the OM stress phenotypes of the *DbamH DbamH2* strain are the same as those of the *DbamH* strain. These data have now been added (lines 284-6 and new ED Fig.9).

The difference between BamH essentiality between different species is indeed an interesting point. Please see our response to the pertinent query by reviewer 4.

- It is interesting that after Ni affinity, SurA is one of the most enriched proteins in the BtBamA pulldowns (Fig. 1b), even though it does not seem to stay bound during the SEC. This finding should be discussed in the light of the recent structures of the BAM-SurA complex (Fenn et al. Nat Commun 15, 7612 (2024); Lehner et al. Sci. Adv. 11, eads6094 (2025)) and perhaps a model of the Bacteroides BAM SurA complexes could be made. We have made the model using AlphaFold and show it below (BtSurA in magenta) for the benefit of the reviewers. BtSurA does appear to occupy a plausible position within the BtBAM complex, but the model confidence is fairly low (ipTM = 0.59). In addition, given that reviewer 1 is clearly less enthusiastic about models that are not supported by experiment, we prefer not to show this in the paper.

- Along the same lines, quantitative proteomics was done after IMAC, whereas the SDS is shown from the SEC. Please show also the proteomics result after SEC.

We feel this is not very useful, but we have now added the gel and proteomics data underlying Figs. 1b and 4b (ED Fig. 1 and Supplementary Table 1), showing quite clearly which co-purifying proteins are contaminants and which ones are not.

- Bt0502 (a 86 kDa beta-barrel protein) in Fig. 1 should be labeled with its normal name rather than with an asterisk. Asterisks are usually used for impurities, but this protein appears to be a regular, virtually stoichiometric component of the complex.

This would indeed seem the case based on Fig. 1a. However, the abundance of Bt0502 in BAM samples is preparation dependent. This is very clear from Fig. 4a, where Bt0502 is much less abundant in the BamA_{his} strain. In addition, our new gel figure (ED Fig. 1) and new proteomics table (Supplementary Table 1) clearly show that Bt0502 (and many others) is a background contaminant as it is also present in wild type. However, we have now labeled it as Bt0502 in Fig. 1a.

Even though it does not appear in the EM map, it must be present on the cryo-EM grids as it stays bound through SEC. This aspect needs further clarification and follow-up characterization.

We have not observed Bt0502 on the cryoEM grids, and together with the points made above, we are confident it is not a component of the BtBAM complex.

One option to get more insight is chemical cross-linking MS to localize intermolecular contacts in the SEC-purified complex and this should definitely be attempted. Could this be a substrate or a structurally variable component? With or without additional data, its structure should be predicted and modeled to the complex, similar to what is done in ED Fig. 10.

As reasoned above, we consider it extremely unlikely Bt0502 is a component of BtBAM. Of course, since Bt0502 is a TBDT, it is a substrate of BtBAM, and therefore would be likely to crosslink with BAM components. It may be more interesting to try and crosslink SLPs to the BAM complex, but considerably effort will be required to account for nonspecific, lipid anchor-driven associations. We hope the reviewers agree that such experiments are beyond the scope of this study.

The predicted model (ipTM 0.56) is interesting in that Bt0502 (below; Bt0502 is magenta) is positioned adjacent to the lateral opening of BamA, similar to SusCD and exactly where it would be expected following folding and OM insertion. However, for the same reasoning as used above for SurA, we have refrained from including this model in the revised manuscript.

- The manuscript contains a long section on proteomics experiments on an BamG deletion strain (lines 317-430). These data are rather preliminary, mostly descriptive, and the resulting conclusions are very weak and often speculative. That part is lengthy and boring to read, and in particular by the contrast to the rest of the manuscript, we find it is not at the impact and interest level of Nature. We recommend to drastically reduce this part to a few key points and put the rest in a supplementary discussion. The unimportance of this part is also evidenced by the fact that it hardly appears in the discussion.

We have shortened this section as requested and added the rest to the supplementary discussion.

- It is suggested (L. 151-152) that the cryo-EM data sets did not contain any other subcomplex populations. Such claim needs to be backed up with a 3D classification and/or variability analysis.

We have included 3D variability analysis results in the new Supplementary Movie 1 and updated the text (lines 140-142) and the cryo-EM data processing workflow in SI Fig. 1a accordingly. Volume reconstructions from three variability modes show heterogeneity in the periplasmic region and extracellular dome of the complex. The 3D variability results are consistent with our original analysis which identified two classes of the BtBAM complex: one with a well-ordered extracellular dome and another one with better-resolved transmembrane and periplasmic regions. We did not observe any major conformational changes or complete disappearance of density corresponding to any of the components, which would have indicated the presence of sub-complexes.

Minor comments:

- ED Fig.2: Interesting Figure, but very hard to see/read, colors appear somewhat aggressive.

We have generated another figure (now ED Fig. 3) with better colours.

- L. 91-92: The result of not being able to knock-out BamF should also be mentioned in the result section – not just in the introduction.

We have now mentioned this (lines 183-4).

- L. 281: The reference to Fig. 4c should probably be Fig. 4d **Indeed. Corrected (line 279).**

- Fig. 4d, bottom left panel: It seem implausible that 0.4 mg/ml deoxycholate is tolerated, but 0.5 mg/ml is completely inhibitory. Could there be some kind of mistake? **This is not a mistake. The data are reproducible even though the exact shape of the curves can be slightly different because of (e.g.) different lag times. An explanation is beyond the scope of the current work.**

- Fig. 4d: After several hours, bacterial growth in delta BamG seems to “restart” (e.g. after 35h, 0.003% SDS). This needs to be commented on. Are these rescue mutations? Can they be sequenced? What is the proteomics of these strains?

This is indeed interesting and could be the result of rescue mutations, and sequencing should be possible. However, given the lack of a functional “smoking gun” resulting from the *DbamH* proteomics we do not think these would be very informative at this stage, and the same could be said about the proteomics. We also note that the peculiar phenotypes could be plate reader-specific, and recapitulating them in larger-scale cultures needed for proteomics may not be possible. We have therefore not commented on this.

- SLAM / Slam is not consistently capitalized in the manuscript

Slam is not capitalised by the group that discovered it (e.g. PMID: 35475756), and we have now used “Slam” consistently.

Referee #3 (Remarks to the Author):

I co-reviewed this manuscript with one of the reviewers who provided the listed reports.

Referee #4 (Remarks to the Author):

Silale, Madej et al. report on the b-barrel assembly machineries or BAM complexes from the Bacteroidota species *B. thetaiotaomicron* and *P. gingivalis*, and show Bacteroidota species to have a distinct BAM composition and architecture, that in addition to core components BamA and BamD, consists of a second b-barrel component (BamF), up to for unique lipoprotein components (BamG, BamH, BamI and BamJ). 3D cryoEM structure of BtBAM and PgBAM reveal the distinct lipoproteins are surface localized rather than periplasmic, and give rise to an extracellular ornament or dome above the BamA subunit.

Deletion of BamG (and to lesser extent BamH, BamI and BamJ) results in growth defects in suboptimal growth media, as well as in an altered OM proteome composition. **Please note that we also observe (limited) growth defects in rich media (Fig. 4c).**

The presence of an extracellular domain on BAM appears to be a unique feature of Bacteroidota BAM complexes, which the authors suggest might be an adaptation to the high complexity of the OM proteome, encompassing a large structural and functional variety in outer membrane β -barrels (OMPs) and associated surface localized lipoproteins (SLPs). The structural features of the BtBAM and PgBAM extracellular dome are suggestive of a role in the processing / folding of client proteins with extensive extracellular decorations, although no formal proof is provided at this stage.

We agree. Note we have toned down speculation on the roles of the new components.

These novel insights in a distinct class of BAM complexes are exciting and will be of interest to the large community studying OM biogenesis in Gram-negative bacteria and/or seeking to target the BAM complex, which is essential for cell viability. Overall, the study is well performed and documented. I have a few suggestions to further clarify the reported data and seek further experimental support for the proposed function of the new BAM components.

We thank the reviewer for their appreciation of our work.

Main points:

- Fig. 1 The affinity pulldown of the Bt BamA His pulldown shows an additional band in Coomassie (Bt0502) and the MS proteomics show additional high likelihood – high fold change proteins (unlabeled). These are not seen in the 3D cryoEM reconstructions suggesting that they are not part of the BtBam complex, or would have dissociated during cryogrid preparation. Can the authors comment on the additional proteins in the MS data? Are these considered contaminants or a possibility of low affinity interactors of the BtBAM complex?

The IMAC pulldowns were done to maximise the detection of potential BAM-interacting proteins via less stringent washing. Therefore, there are many hits that have clearly nothing to do with BAM, including abundant cytoplasmic proteins such as large ribosomal subunits. As shown in new Supplementary Table 1 (and ED Fig. 1), most of the hits are contaminants since they are also present in the WT sample. Some hits, e.g. Bt4261 and Bt3222 (both SLPs) may have functional relevance since they are relatively abundant and present in both *bamA-his* (Fig. 1b) and *bamA-his DbamHhis* (Fig. 4b). Evaluation of their potential role(s) in BAM function will require further work.

The Bt0502 band appears to be missing from the BtBamA his pulldown in Figure 4a. Were purification conditions different?

The band isn't missing but weak (and now labeled). The amount of co-purifying contaminants such as Bt0502 is indeed preparation dependent, e.g. due to different pooling of SEC fractions or different efficiency of the IMAC.

Similarly, in Figure 5, what are the high logP / high fold change proteins not labelled or colored (i.e. grey)?

We have added the functional annotations to the corresponding supplementary tables 4 and 5. Since these are total membrane proteomes (total cell proteomes for *P. gingivalis*), there are many more hits than from IMAC, originating from every cellular compartment. We think this underscores the far-reaching (indirect) consequences of removal of the extracellular dome. Please note that the *bamH* deletion doesn't affect BamA and BamG abundance (Fig. 4a and Supplementary Table 1), suggesting that we have uncoupled the effects resulting from extracellular dome removal from those caused by defects in the BamADG core complex.

- The Berks and Lea lab reported the isolation of the BAM complex of the Bacteroidota species *B. johnsoniae* (<https://www.biorxiv.org/content/10.1101/2025.02.17.638638v1>), identifying BamF and BamG as new core components of the Bj-BAM complex, as well as non-essential subunits BamM and BamP. BamM (which is also containing a PPI domain) appears to take a similar position to BamH alongside BamG, creating a large extracellular ornament to the BAM complex. In the discussion, the authors should comment on the possible homology or analogy of BamH and BamM. In case homologues, is there a reason to differentiate between BamH in *Pg* and *Bt* on the one hand and BamM in *Bj*, rather than harmonizing subunit names over the two studies?

Given that the papers won't be published back-to-back we have not (yet) addressed this issue but might address it during the proofreading stage.

As suggested by Fig. 3a in the Liu et al. paper, BamM has a very narrow distribution, and there is no appreciable homology (sequence or structural) between BamI and BamM (please note the revised naming in response to reviewers 1 and 2, but this doesn't affect BamM and BamP). BamI does not have a PPI domain and does not bind calcium ions, and is therefore clearly different from BamM.

Same for BamP, is there any indication of candidate BamP homologues in *P. gingivalis* and/or *B. thetaiotaomicron*?

As shown in Fig. 3a from Liu et al., BamP, like BamM, has a very narrow distribution and is confined to the family Flavobacteriaceae. The differences in subunit composition highlight that the BAM complexes from *F. johnsoniae* and those from *B. theta* and *P. gingivalis* differ substantially.

- In Liu et al. (<https://www.biorxiv.org/content/10.1101/2025.02.17.638638v1>), BamG could only be removed in *B. johnsoniae* in the form of a depletion mutant or in

presence of a suppressor mutation of BamA. Can the authors exclude a suppressor mutation in their knockout mutants in Bt and Pg? Given this discrepancy, it would be advisable to sequence the mutants used in this study.

We thank the reviewer for raising this important point. We have sequenced *bamA-his DbamH* and the parental *bamA-his* strain. We find no suppressor mutations in the *DbamH* strain (lines 334-337), again underscoring the differences between the BAM complexes from *F. johnsoniae* and those from *B. theta* and *P. gingivalis*.

- In Ext. Data Figure 10, the authors present an alphafold3 model for BtBAM in complex with SusCD_lev. What is the value of this addition? Without an experimental validation, any conclusions based on this model are rather speculative (Ln 490).

We did not draw conclusions from this figure. Rather, we suggested "that this structure may form an extracellular chaperone or assembly cage for the formation of SLP-barrel complexes that are a hallmark of the Bacteroidetes phylum. This hypothesis is supported by the fact that the cage is large enough to accommodate folded SusCD complexes (ED Fig. 10)".

However, we agree this is speculative and, also taking into account reviewer 1, we have now removed the figure.

The presence of an extensive extracellular dome on BtBAM and PgBAM is suggestive of an involvement in substrate processing (folding) at the cell surface, but there is no unambiguous data in the present study to claim a functional assignment to the dome. The authors now use proteinase K to assess surface exposure of BAM substrates (SLPs and OMPs). They could consider using limited proteolysis assays to assess the folding state of the presumed BAM clients.

Unfortunately, we're not sure what the reviewer wants us to do. Limited proteolysis of purified OMPs or whole cells?

While nothing is known about OMP quality control in Bacteroidota, it stands to reason that sophisticated systems are present to remove wrongly folded barrels and SLPs. Thus, purified OMPs are likely to be correctly folded. Limited proteolysis of whole cells might be informative, but it is possible (and perhaps likely) that only correctly folded proteins make it to the cell surface. In addition, given the lack of a functional "smoking gun" from the *DbamH* proteomics we do not think these experiments would be a game-changer at this point.

Minor points:

- Ln 140-141 The authors mention Alphafold2 (AF2) models to dock unambiguously into the cryoEM density. Please provide CC values for their fits with the density. AF2 models were docked manually.

- Bacteroidetes was recently renamed to Bacteroidota (<https://ncbiinsights.ncbi.nlm.nih.gov/2021/12/10/ncbi-taxonomy-prokaryote-phylaadded/>). It would be advisable to use the new nomenclature.

Addressed as requested throughout the manuscript.